# NF1 regulates mesenchymal glioblastoma plasticity and aggressiveness through the AP-1 transcription factor FOSL1

**Carolina Marques[1], Thomas Unterkircher[2], Paula Kroon[1], Barbara Oldrini[1], Annalisa Izzo[2], Yuliia Dramaretska[3], Roberto Ferrarese[2], Eva Kling[2], Oliver Schnell[2], Sven Nelander[4,5], Erwin F Wagner[6,7,8], Latifa Bakiri[6,7], Gaetano Gargiulo[3], Maria Stella Carro[2]\*, Massimo Squatrito[1]\***

[1]Seve Ballesteros Foundation Brain Tumor Group, Spanish National Cancer Research Centre, Madrid, Spain; [2]Department of Neurosurgery, Faculty of Medicine Freiburg, Freiburg, Germany; [3]Max-Delbrück-Center for Molecular Medicine in the Helmholtz Association (MDC), Berlin, Germany; [4]Dept of Immunology, Genetics and Pathology and Science for Life Laboratory, Uppsala University, Rudbecklaboratoriet, Uppsala, Sweden; [5]Science for Life Laboratory, Uppsala University, Rudbecklaboratoriet, Uppsala, Sweden; [6]Genes, Development, and Disease Group, Spanish National Cancer Research Centre, Madrid, Spain; [7]Laboratory Medicine Department, Medical University of Vienna, Vienna, Austria; [8]Dermatology Department, Medical University of Vienna, Vienna, Austria

**Abstract** The molecular basis underlying glioblastoma (GBM) heterogeneity and plasticity is not fully understood. Using transcriptomic data of human patient-derived brain tumor stem cell lines (BTSCs), classified based on GBM-intrinsic signatures, we identify the AP-1 transcription factor *FOSL1* as a key regulator of the mesenchymal (MES) subtype. We provide a mechanistic basis to the role of the neurofibromatosis type 1 gene (*NF1*), a negative regulator of the RAS/MAPK pathway, in GBM mesenchymal transformation through the modulation of *FOSL1* expression. Depletion of *FOSL1* in *NF1*-mutant human BTSCs and *Kras*-mutant mouse neural stem cells results in loss of the mesenchymal gene signature and reduction in stem cell properties and in vivo tumorigenic potential. Our data demonstrate that *FOSL1* controls GBM plasticity and aggressiveness in response to *NF1* alterations.

**\*For correspondence:**
Correspondence: maria.carro@uniklinik-freiburg.de; msquatrito@cnio.es

**Competing interests:** The authors declare that no competing interests exist.

## Introduction

Gliomas are the most common primary brain tumor in adults. Given the strong association of the *isocitrate dehydrogenase 1* and *2* (*IDH1/2*) genes mutations with glioma patients survival, the 2016 WHO classification, which integrates both histological and molecular features, has introduced the distinction of *IDH*-wildtype (IDH-wt) or *IDH*-mutant (IDH-mut) in diffuse gliomas (*Louis et al., 2016*). IDH-wt glioblastoma (GBM) represents the most frequent and aggressive form of gliomas, characterized by high molecular and cellular inter- and intra-tumoral heterogeneity.

Large-scale sequencing approaches have evidenced how concurrent perturbations of cell cycle regulators, growth and survival pathways, mediated by RAS/MAPK and PI3K/AKT signaling, play a significant role in driving adult GBMs (*Brennan et al., 2013*; *Cancer Genome Atlas Research Network, 2008*; *Verhaak et al., 2010*). Moreover, various studies have classified GBM in different subtypes, using transcriptional profiling, being now the proneural (PN), classical (CL), and mesenchymal (MES) the most widely accepted (*Phillips et al., 2006*; *Verhaak et al., 2010*; *Wang et al., 2017*).

Patients with the MES subtype tend to have worse survival rates compared to other subtypes, both in the primary and recurrent tumor settings (*Wang et al., 2017*). The most frequent genetic alterations – neurofibromatosis type 1 gene (*NF1*) copy number loss or mutation – and important regulators of the MES subtype, such as *STAT3*, *CEBPB*, and *TAZ*, have been identified (*Bhat et al., 2011*; *Carro et al., 2010*; *Verhaak et al., 2010*). Nevertheless, the mechanisms of regulation of MES GBMs are still not fully understood. For example, whether the MES transcriptional signature is controlled through tumor cell-intrinsic mechanisms or influenced by the tumor microenvironment (TME) is still an unsolved question. In fact, the critical contribution of the TME adds another layer of complexity to MES GBMs. Tumors from this subtype are highly infiltrated by non-neoplastic cells as compared to PN and CL subtypes (*Wang et al., 2017*). Additionally, MES tumors express high levels of angiogenic markers and exhibit high levels of necrosis (*Cooper et al., 2012*).

Even though each subtype is associated with specific genetic alterations, there is a considerable plasticity among them: different subtypes coexist in the same tumors and shifts in subtypes can occur over time (*Patel et al., 2014*; *Sottoriva et al., 2013*). This plasticity may be explained by acquisition of new genetic and epigenetic abnormalities, stem-like reprogramming, or clonal variation (*Fedele et al., 2019*). It is also not fully understood whether the distinct subtypes evolve from a common glioma precursor (*Ozawa et al., 2014*). For instance, PN and CL tumors often switch phenotype to MES upon recurrence, and treatment also increases the mesenchymal gene signature, suggesting that MES transition, or epithelial to mesenchymal (EMT)-like, in GBM is associated with tumor progression and therapy resistance (*Bhat et al., 2013*; *Halliday et al., 2014*; *Phillips et al., 2006*). Yet, the frequency and relevance of this EMT-like phenomenon in glioma progression remains unclear. EMT has also been associated with stemness in other cancers (*Mani et al., 2008*; *Tam and Weinberg, 2013*; *Ye et al., 2015*). Glioma stem cells (GSCs) share features with normal neural stem cells (NSCs) such as self-renewal and ability to differentiate into distinct cellular lineages (astrocytes, oligodendrocytes, and neurons) but are thought to be responsible for tumor relapse, given their ability to repopulate tumors and their resistance to treatment (*Bao et al., 2006*; *Chen et al., 2012*). GSCs heterogeneity is also being increasingly observed (*Bhat et al., 2013*; *Mack et al., 2019*; *Richards et al., 2021*), but whether genotype-to-phenotype connections exist remain to be clarified.

*FOSL1*, which encodes FRA-1, is an AP-1 transcription factor (TF) with prognostic value in different epithelial tumors, where its overexpression correlates with tumor progression or worse patient survival (*Chiappetta et al., 2007*; *Gao et al., 2017*; *Usui et al., 2012*; *Vallejo et al., 2017*; *Wu et al., 2015*; *Xu et al., 2017*). Moreover, the role of *FOSL1* in EMT has been documented in breast and colorectal cancers (*Andreolas et al., 2008*; *Bakiri et al., 2015*; *Diesch et al., 2014*). In GBM, it has been shown that *FOSL1* modulates in vitro glioma cell malignancy (*Debinski and Gibo, 2005*).

Here we report that *NF1* loss, by increasing RAS/MAPK activity, modulates *FOSL1* expression, which in turn plays a central function in the regulation of MES GBM. Using a surrogate mouse model of MES GBM and patient-derived MES brain tumor stem cells (BTSCs), we show that *FOSL1* is responsible for sustaining cell growth in vitro and in vivo, and for the maintenance of stem-like properties. We propose that *FOSL1* is an important regulator of GBM stemness, MES features and plasticity, controlling an EMT-like process with therapeutically relevant implications.

## Results

### *FOSL1* is a key regulator of the MES subtype

To study the tumor cell-intrinsic signaling pathways that modulate the GBM expression subtypes, we assembled a collection of transcriptomic data (both expression arrays and RNA-sequencing) of 144 samples derived from 116 independent BTSC lines (see Materials and methods for details). Samples were then classified according to the previously reported 50-gene glioma-intrinsic transcriptional subtype signatures and the single-sample gene set enrichment analysis (ssGSEA)-based equivalent distribution resampling classification strategy (*Wang et al., 2017*). Principal component analysis (PCA) showed a large overlap of the transcription profile among BTSCs classified either as CL/PN while most of the MES appeared as separate groups (*Figure 1A* and *Supplementary file 1*). This separation is consistent with early evidence in GSCs (*Bhat et al., 2013*) and holds 92% of concordance in the identification of a recent two transcriptional subgroups classification of single-GSCs

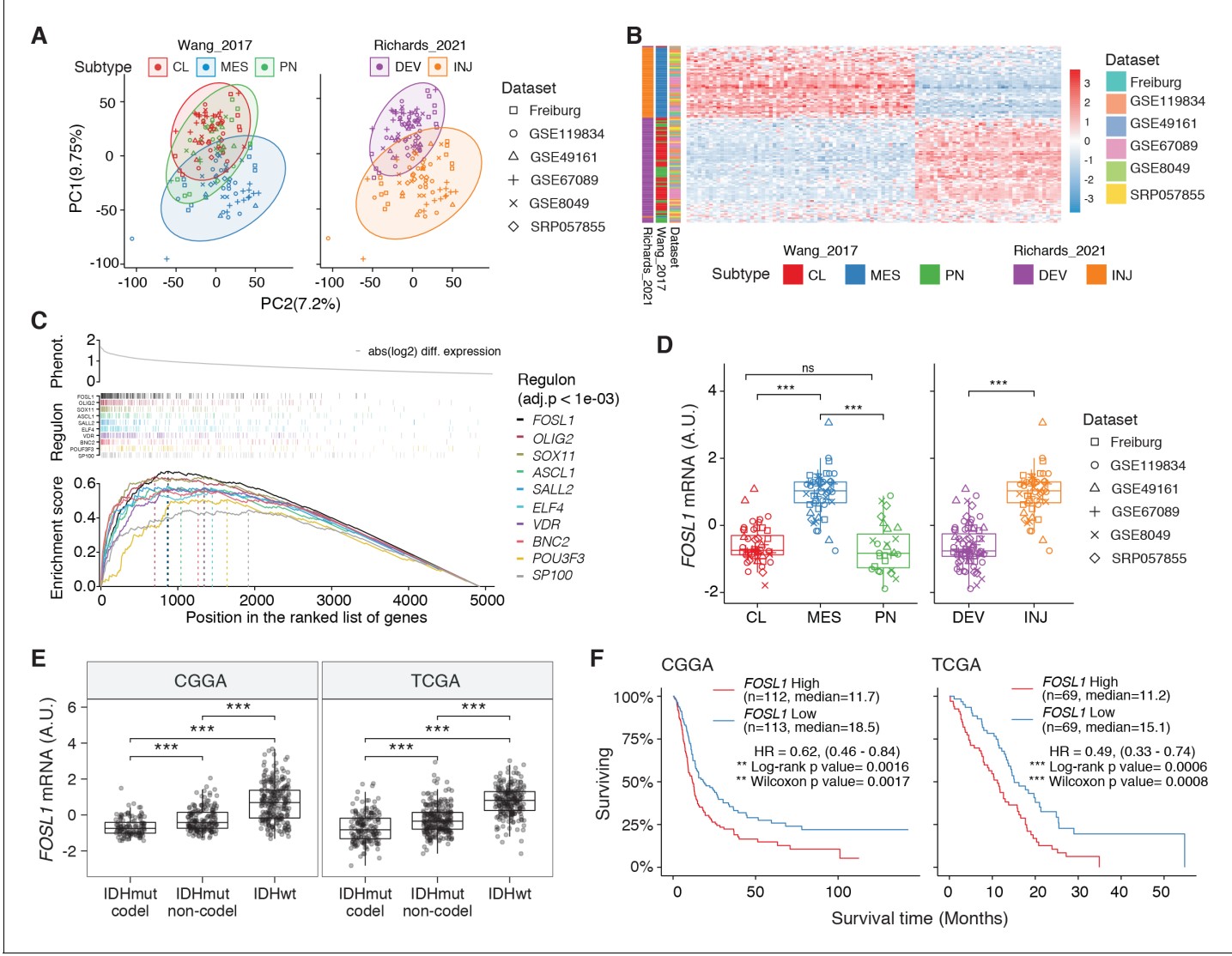

**Figure 1.** *FOSL1* is a *bona fide* regulator of the glioma-intrinsic mesenchymal (MES) transcriptional signature. (**A**) Principal component (PC) analysis of the brain tumor stem cells (BTSCs) expression dataset. (**B**) Heatmap of the top 100 differentially expressed genes between MES and non-MES BTSCs. (**C**) One-tail gene set enrichment analysis (GSEA) of the top 10 scoring transcription factors (TFs) in the master regulator analysis (MRA). (**D**) *FOSL1* mRNA expression in the BTSCs dataset. One-way ANOVA with Tukey multiple pairwise comparison, \*\*\*p≤0.001, ns = not significant. (**E**) *FOSL1* mRNA expression in the CGGA and TCGA datasets. Tumors were separated according to their molecular subtype classification. One-way ANOVA with Tukey multiple pairwise comparison, \*\*\*p≤0.001. (**F**) Kaplan–Meier survival curves of IDH-wt gliomas in the CGGA and TCGA datasets stratified based on *FOSL1* expression (see Materials and methods for details).

The online version of this article includes the following source data and figure supplement(s) for figure 1:

**Source data 1.** Source data of *Figure 1A, B, D–*F.

**Figure supplement 1.** Expression, bulk and single-cell RNA-seq, of the top 10 transcription factors (TFs) identified in the master regulator analysis (MRA).

**Figure supplement 1—source data 1.** Source data of *Figure 1—figure supplement 1A, C, D, and E*.

**Figure supplement 2.** Expression in human glioblastomas (GBMs) of *FOSL1* and the top 10 transcription factors (TFs) identified in the master regulator analysis (MRA).

**Figure supplement 2—source data 1.** Source data of *Figure 1—figure supplement 2A–C*.

defined as developmental (DEV) and injury response (INJ) (*Richards et al., 2021*). Differential gene expression analysis comparing mesenchymal versus non-mesenchymal BTSCs confirmed the clear separation among the two groups, with only a minor fraction of cell lines showing a mixed

expression profile (*Figure 1B* and *Supplementary file 2*), further supporting that GSCs exist along a major transcriptional gradient between two cellular states (*Bhat et al., 2013*; *Richards et al., 2021*).

To reveal the signaling pathways underlying the differences between MES and non-MES BTSCs, we then applied a network-based approach based on the Algorithm for the Reconstruction of Accurate Cellular Networks (ARACNe) (*Basso et al., 2005*; *Carro et al., 2010*), which identifies a list of TFs with their predicted targets, defined as regulons. The regulon for each TF is constituted by all the genes whose expression data exhibit significant mutual information with that of a given TF and are thus expected to be regulated by that TF (*Castro et al., 2016*; *Fletcher et al., 2013*). Enrichment of a relevant gene signature in each of the regulons can point to the TFs acting as master regulators (MRs) of the response or phenotype (*Carro et al., 2010*; *Fletcher et al., 2013*). Master regulator analysis (MRA) identified a series of TFs, among which *FOSL1*, *VDR*, *OLIG2*, *SP100*, *ELF4*, *SOX11*, *BNC2*, *ASCL1*, *SALL2*, and *POU3F3* were the top 10 most statistically significant (Benjamini–Hochberg p<0.0001) (*Figure 1C* and *Supplementary file 3*). *FOSL1*, *VDR*, *SP100*, *ELF4*, and *BNC2* were significantly upregulated in the MES BTSCs, while *OLIG2*, *SOX11*, *ASCL1*, *SALL2*, and *POU3F3* were upregulated in the non-MES BTSCs (*Figure 1D* and *Figure 1—figure supplement 1A*). Gene set enrichment analysis (GSEA) evidenced how the regulons for the top 10 TFs are enriched for genes that are differentially expressed among the two classes (MES and non-MES) with *FOSL1* having the highest enrichment score (*Figure 1C*, *Figure 1—figure supplement 1B*, and *Supplementary file 3*). Lastly, an analysis of an independent BTSCs dataset (*Richards et al., 2021*) evidenced that the differential expression of *FOSL1* and the other TFs was maintained both at bulk (*Figure 1—figure supplement 1C*) and at a single-cell level (*Figure 1—figure supplement 1D, E*).

We then analyzed the CGGA and TCGA pan-glioma datasets (*Ceccarelli et al., 2016*; *Zhao et al., 2017*) and observed that *FOSL1* expression is elevated in the IDH-wt glioma molecular subgroup (*Figure 1E* and *Supplementary file 4*) with a significant upregulation in the MES subtype in bulk tumors, and it is also enriched in MES-like cells (*Neftel et al., 2019*) at the single-cell level (*Figure 1—figure supplement 2A–C*). Importantly, high expression levels were associated with worse prognosis in IDH-wt tumors (*Figure 1F*), thus suggesting that *FOSL1* could represent not only a key regulator of the glioma-intrinsic MES signature, but also a putative key player in MES glioma pathogenesis.

## *NF1* modulates the MES signature and *FOSL1* expression

*NF1* alterations and activation of the RAS/MAPK signaling have been previously associated with the MES GBM subtype (*Brennan et al., 2013*; *Verhaak et al., 2010*; *Wang et al., 2016*; *Wang et al., 2017*). However, whether *NF1* plays a broader functional role in the regulation of the MES gene signature (MGS) in IDH-wt gliomas still remains to be established.

We initially grouped, according to the previously described GBM subtype-specific gene signatures, a subset of IDH-wt glioma samples of the TCGA dataset for which RNA-seq data were available (n = 229) (see Materials and methods for details). By analyzing the frequency of *NF1* alterations (either point mutations or biallelic gene loss), we confirmed a significant enrichment of *NF1* alterations in MES versus non-MES tumors (Fisher's exact test p=0.0106) (*Figure 2A, B*). Importantly, we detected higher level of *FOSL1* mRNA in the cohort of IDH-wt gliomas with *NF1* alterations (Student's t test p=0.018) (*Figure 2C*), as well as a significant negative correlation between *FOSL1* and *NF1* mRNA levels (Pearson R = −0.44, p=7.8e-12) (*Figure 2D* and *Supplementary file 4*).

To test whether a NF1-MAPK signaling is involved in the regulation of *FOSL1* and the MES subtype, we manipulated *NF1* expression in patient-derived GBM tumorspheres of either MES or non-MES subtypes. To recapitulate the activity of the full-length NF1 protein, we transduced the cells with the NF1 GTPase-activating domain (NF1-GRD), spanning the whole predicted Ras GTPase-activating (GAP) domain (*McCormick, 1990*). NF1-GRD expression in the MES cell line BTSC 233 led to (i) inhibition of RAS activity as confirmed by analysis of pERK expression upon EGF or serum stimulation (*Figure 2—figure supplement 1A, B*) as well as by RAS pull-down assay (*Figure 2—figure supplement 1C*); (ii) strong reduction of a RAS-induced oncogenic signature expression (NES = −1.7; FDR q-value < 0.001) (*Figure 2—figure supplement 1D*); and (iii) diminished cell proliferation (*Figure 2—figure supplement 1E, F*). Consistent with the negative correlation of *FOSL1* and *NF1* mRNA levels in IDH-wt gliomas (*Figure 2D*), NF1-GRD overexpression in two independent MES GBM lines (BTSC 233 and BTSC 232) was associated with a significative downregulation of *FOSL1* and *FOSL1*-regulated genes (*Figure 2E* and *Figure 2—figure supplement 2A–C*). Concurrently, we

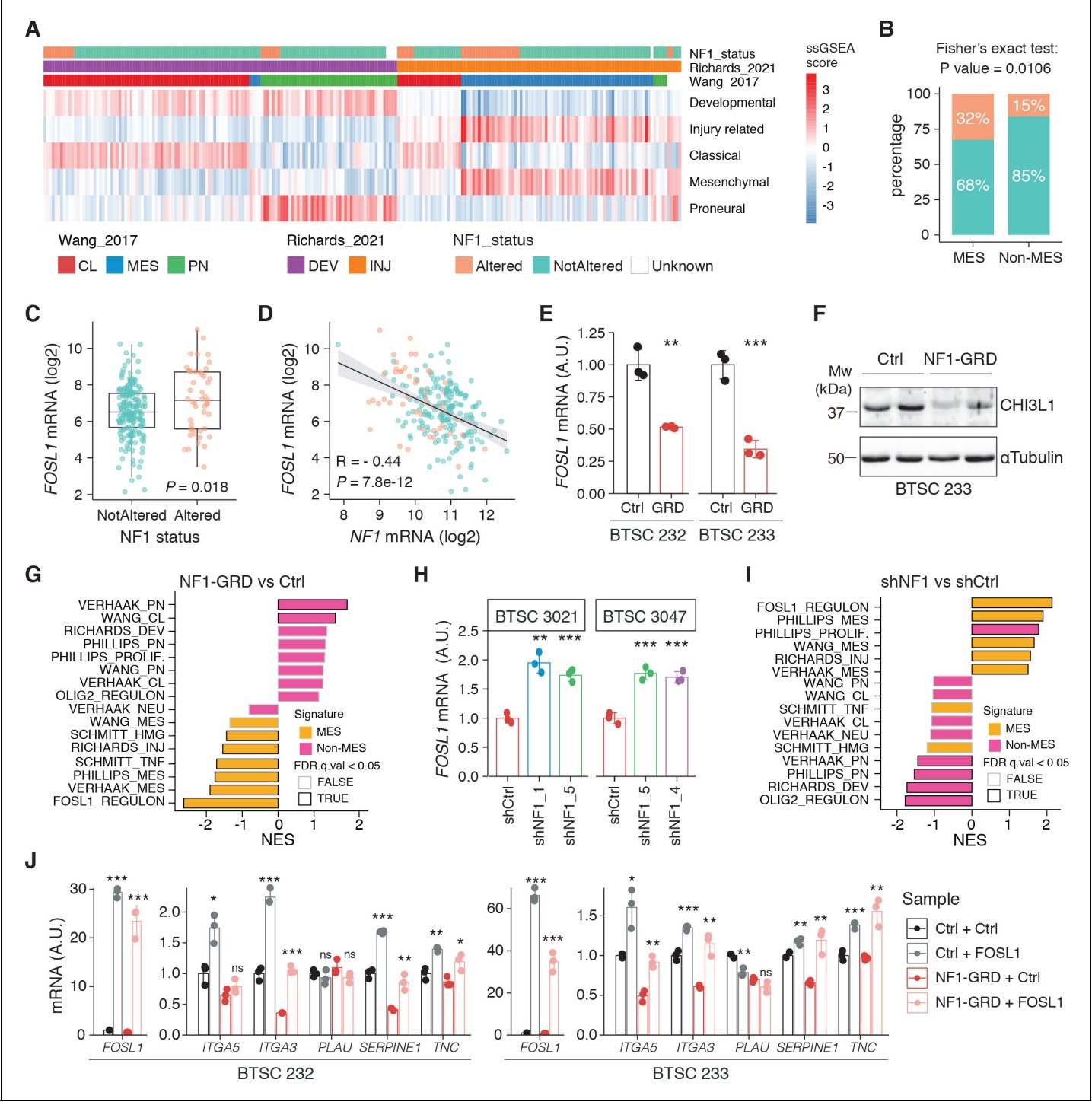

**Figure 2.** *NF1* is a functional modulator of mesenchymal (MES) transcriptional signatures through *FOSL1* expression regulation. (A) Heatmap of the subtypes single-sample gene set enrichment analysis (ssGSEA) scores and *NF1* genetic alterations of the IDH-wt gliomas in the TCGA dataset. (B) Frequency of *NF1* alterations in MES and non-MES IDH-wt gliomas. Colors are as in panel (A). (C) *FOSL1* mRNA expression in IDH-wt gliomas, stratified according to *NF1* alterations. Colors are as in panel (A). Student's t test, p=0.018. (D) Correlation of *FOSL1* and *NF1* mRNA expression in IDH-wt gliomas. Colors are as in panel (A). Pearson correlation, R = −0.044, p=7.8e-12. (E) qRT-PCR analysis of *FOSL1* expression upon NF1-GRD overexpression in BTSC 232 and BTSC 233 cells. (F) Western blot analysis of whole-cell extract of BTSC 233 cells showing CHI3L1 mesenchymal marker expression upon NF1-GRD transduction; α-tubulin was used as loading control. Two biological replicates are shown. (G) Gene set enrichment analysis (GSEA) results of BTSC 233 cells transduced with NF1-GRD expressing lentivirus versus Ctrl. NES: normalized enrichment score. (H) qRT-PCR analysis of *FOSL1* expression upon *NF1* knockdown in BTSC 3021 and BTSC 3047 cells. (I) GSEA results of BTSC 3021 transduced with sh*NF1*_5 versus

*Figure 2 continued on next page*

*Figure 2 continued*

Ctrl. (J) qRT-PCR analysis of MES genes expression upon NF1-GRD and *FOSL1* co-expression in BTSC 232 and BTSC 233 cells. qRT-PCR data in (E), (H), and (J) are presented as mean ± SD (n = 3, technical replicates), normalized to 18S rRNA expression; Student's t test, *p≤0.05, **p≤0.01, ***p≤0.001, ns = not significant.

The online version of this article includes the following source data and figure supplement(s) for figure 2:

**Source data 1.** Source data of *Figure 2F*.
**Source data 2.** Source data of *Figure 2A, C–E, G–J*.
**Figure supplement 1.** NF1-GRD expression leads to downregulation of RAS signaling.
**Figure supplement 1—source data 1.** Source data of *Figure 2—figure supplement 1A*.
**Figure supplement 1—source data 2.** Source data of *Figure 2—figure supplement 1C*.
**Figure supplement 2.** Modulation of *NF1* expression regulates *FOSL1* targets and mesenchymal genes.
**Figure supplement 2—source data 1.** Source data of *Figure 2—figure supplement 2A*.
**Figure supplement 2—source data 2.** Source data of *Figure 2—figure supplement 2E*.
**Figure supplement 2—source data 3.** Source data of *Figure 2—figure supplement 2J*.
**Figure supplement 2—source data 4.** Source data of *Figure 2—figure supplement 2C, G–I*.
**Figure supplement 3.** MAPK inhibition reverts the effects of *NF1* silencing on *FOSL1* and mesenchymal genes expression.
**Figure supplement 3—source data 1.** Source data of *Figure 2—figure supplement 3A*.
**Figure supplement 3—source data 2.** Source data of *Figure 2—figure supplement 3C*.
**Figure supplement 3—source data 3.** Source data of *Figure 2—figure supplement 3E*.
**Figure supplement 3—source data 4.** Source data of *Figure 2—figure supplement 3B, D, F, and G*.

also observed a significant decrease of two well-characterized mesenchymal features, namely CHI3L1 expression (*Figure 2F*) as well as the ability of MES GBM cells to differentiate into osteocytes, a feature shared with mesenchymal stem cells (*Ricci-Vitiani et al., 2008*; *Tso et al., 2006*; *Figure 2—figure supplement 2D*). Moreover, NF1-GRD expression led to a significant reduction of the *FOSL1* regulon and the MGSs, with a concurrent increase of the *OLIG2* regulon and the non-MES gene signatures (non-MGSs) (*Figure 2G*).

Conversely, *NF1* knockdown with three independent shRNAs (shNF1_1, shNF1_4, and shNF1_5) in two non-MES lines (BTSC 3021 and BTSC 3047) (*Figure 2—figure supplement 2E*) led to an upregulation of *FOSL1* (*Figure 2H*), with a concomitant significant increase in its targets (*Figure 2—figure supplement 2F, G*), an upregulation of the MGSs, and downregulation of the N-MGSs (*Figure 2I*).

The observed NF1-mediated gene expression changes might be potentially driven by an effect on *FOSL1* or other previously described mesenchymal TFs (such as *BHLHB2, CEBPB, FOSL2, RUNX1, STAT3,* and *TAZ;*) (*Bhat et al., 2011*; *Carro et al., 2010*). Interestingly, only *FOSL1,* and to some extent *CEBPB,* was consistently downregulated upon NF1-GRD expression (*Figure 2—figure supplement 2H*) and upregulated following *NF1* knockdown (*Figure 2—figure supplement 2I*). To then test whether *FOSL1* was playing a direct role in the *NF1*-mediated regulation of mesenchymal genes expression, we overexpressed *FOSL1* in the MES GBM lines transduced with the NF1-GRD (*Figure 2—figure supplement 2J*). qRT-PCR analysis showed that *FOSL1* was able to rescue the NF1-GRD-mediated downregulation of mesenchymal genes, such as *ITGA3, ITGA5, SERPINE1,* and *TNC* (*Figure 3J*). Lastly, exposure of *NF1* silenced cells to the MEK inhibitor GDC-0623, led to a reduction of *FOSL1* upregulation, both at the protein and the mRNA levels, as well as to a downregulation of the mesenchymal genes *ITGA3* and *SERPINE1* (*Figure 2—figure supplement 3A, B*).

Overall these evidences implicate the NF1-MAPK signaling in the regulation of the MGSs through the modulation of *FOSL1* expression.

## *Fosl1* deletion induces a shift from a MES to a PN gene signature

To further explore the NF1-MAPK-FOSL1 axis in MES GBM, we used a combination of the RCAS-Tva system with the CRISPR/Cas9 technology, recently developed in our laboratory (*Oldrini et al., 2018*), to induce *Nf1* loss or *Kras* mutation. Mouse NSCs from *hGFAP-Tva; hGFAP-Cre; Trp53^lox; ROSA26-LSL-Cas9* pups were isolated and infected with viruses produced by DF1 packaging cells transduced with RCAS vectors targeting the expression of *Nf1* through shRNA and sgRNA (sh*Nf1* and sg*Nf1*) or overexpressing a mutant form of *Kras* (*Kras^{G12V}*). Loss of NF1 expression was confirmed by western blot, and FRA-1 was upregulated in the two models of *Nf1* loss compared to

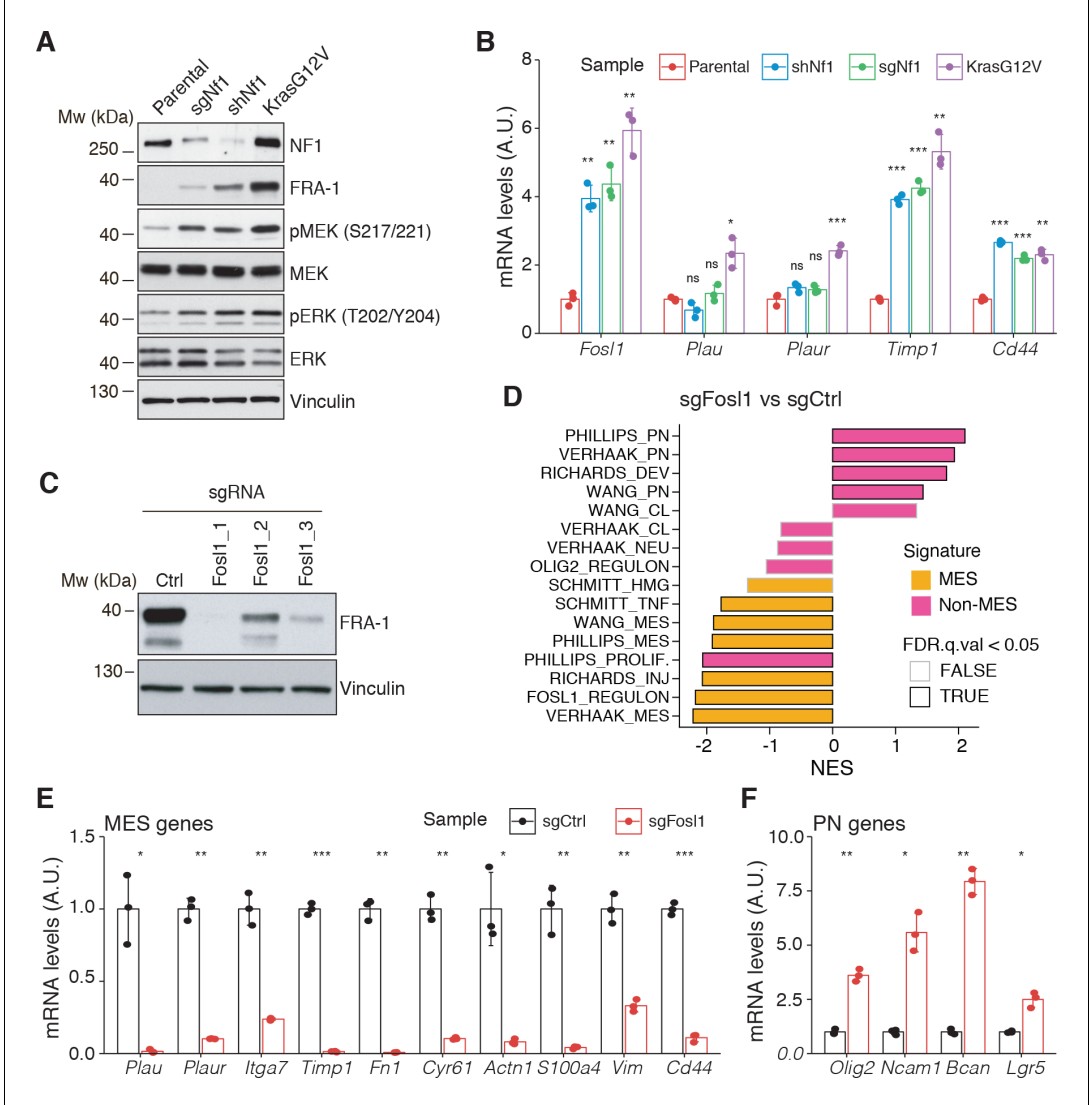

**Figure 3.** *Fosl1* is induced by MAPK kinase activation and is required for mesenchymal (MES) gene expression. (**A**) Western blot analysis using the specified antibodies of p53-null neural stem cells (NSCs), parental and infected with sg*Nf1*, sh*Nf1*, and *Kras*$^{G12V}$; vinculin was used as loading control. (**B**) mRNA expression of *Fosl1* and MES genes (*Plau, Plaur, Timp1,* and *Cd44*) in infected p53-null NSCs compared to parental cells (not infected). Data from a representative of two experiments are presented as mean ± SD (n = 3), normalized to *Gapdh* expression. Student's t test, relative to parental cells: ns = not significant, *p≤0.05, **p≤0.01, ***p≤0.001. (**C**) FRA-1 expression detected by western blot in p53-null *Kras*$^{G12V}$ NSCs upon transduction with sgRNAs targeting *Fosl1*, after selection with 1 µg/mL puromycin; vinculin was used as loading control. (**D**) Gene set enrichment analysis (GSEA) results of p53-null *Kras*$^{G12V}$ sg*Fosl1*_1 versus sgCtrl NSCs. (**E, F**) mRNA expression of MES (**E**) and PN genes (**F**) in sgCtrl and sg*Fosl1*_1 p53-null *Kras*$^{G12V}$ NSCs. Data from a representative of two experiments are presented as mean ± SD (n = 3, technical replicates), normalized to *Gapdh* expression. Student's t test, relative to sgCtrl: *p≤0.05; **p≤0.01; ***p≤0.001.

The online version of this article includes the following source data for figure 3:

**Source data 1.** Source data of *Figure 3A*.
**Source data 2.** Source data of *Figure 3C*.
**Source data 3.** Source data of *Figure 3B, D–F*.

parental cells and further upregulated in cells infected with *Kras*$^{G12V}$ (**Figure 3A**). Consistent with activation of the Ras signaling, as a result of both *Nf1* loss and *Kras* mutation, the MEK/ERK pathway was more active in infected cells compared to parental cells (**Figure 3A**). Higher levels of activation of the MEK/ERK pathway were more pronounced in the *Kras* mutant cells and were associated with a stronger induction of mesenchymal genes such as *Plau, Plaur, Timp1,* and *Cd44* (**Figure 3B**).

Moreover, the upregulation of both FRA-1 and the mesenchymal genes was blocked by exposing sh*Nf1* and *Kras* mutant cells to the MAPK inhibitors trametinib or U0126 (*Figure 2—figure supplement 3C, D*).

Taking advantage of the Cas9 expression in the generated p53-null NSCs models, *Fosl1* was knocked out through sgRNAs. Efficient downregulation of FRA-1 was achieved with two different sgRNAs (*Figure 3C* and *Figure 2—figure supplement 3E*). Cells transduced with sg*Fosl1*_1 and sg*Fosl1*_3 were then subjected to further studies.

As suggested by the data presented here on the human BTSCs datasets and cell lines, *FOSL1* appears to be a key regulator of the MES subtype. Consistently, RNA-seq analysis followed by GSEA of p53-null *Kras*$^{G12V}$ sg*Fosl1*_1 versus sgCtrl revealed a significant loss of the MGSs and increase in the N-MGSs (*Figure 3D*). These findings were validated by qRT-PCR with a significant decrease in expression of a panel of MES genes (*Plau*, *Itga7*, *Timp1*, *Plaur*, *Fn1*, *Cyr61*, *Actn1*, *S100a4*, *Vim*, *Cd44*) (*Figure 3E*) and increased expression of PN genes (*Olig2*, *Ncam1*, *Bcan*, *Lgr5*) in the *Fosl1* knock-out (KO) *Kras*$^{G12V}$ NSCs (*Figure 3F*). A similar trend was observed in the *Fosl1* KO sh*Nf1* NSCs (*Figure 2—figure supplement 3F, G*), and the extent of MSG regulation appeared proportional to the extent of MAPK activation by individual perturbations (*Figure 3A*).

Altogether, these data indicated that *Kras*$^{G12V}$–transduced cells, which show the highest *FOSL1* expression and mesenchymal commitment, are a suitable model to functionally study the role of a MAPK-FOSL1 axis in MES GBM.

## *Fosl1* depletion affects the chromatin accessibility of the mesenchymal transcription program and differentiation genes

FOSL1 is a member of the AP-1 TF super family, which may be composed of a diverse set of homo- and heterodimers of the individual members of the JUN, FOS, ATF, and MAF protein families. In GBM, AP-1 can act as a pioneer factor for other transcriptional regulators, such as ATF3, to coordinate response to stress in GSCs (*Gargiulo et al., 2013*). To test the effect of *Fosl1* ablation on chromatin regulation, we performed open chromatin profiling using ATAC-seq in the p53-null *Kras*$^{G12V}$ NSCs model (*Figure 3C*). This analysis revealed that *Fosl1* loss strongly affects chromatin accessibility of known cis-regulatory elements such as transcription start sites (TSS) and CpG islands (CGI), as gauged by unsupervised clustering of *Fosl1* wild-type and KO cells (*Figure 4A*). Consistent with a role for *FOSL1*/FRA-1 in maintaining chromatin accessibility at direct target genes, deletion of *Fosl1* caused the selective closing of chromatin associated with the major AP-1 TFs binding sites (*Figure 4B*). Upon *Fosl1* loss, profiling of the motifs indicated that chromatin associated with AP-1/2 TFs binding were closed and – conversely – a diverse set of general and lineage-specific TFs, including MFZ1, NRF1, RREB1, and others (*Figure 4C*), were opened. The genes associated with changes in chromatin accessibility upon *Fosl1* loss are involved in several cell fate commitment, differentiation, and morphogenesis programs (*Figure 4D, E*). Next, we investigated chromatin remodeling dynamics using *limma* and identified 9749 regions with significant differential accessibility (absolute log2 fold-change >1, FDR < 0.05). Importantly, *Fosl1* loss induced opening of chromatin associated with lineage-specific markers, along with closing of chromatin at the *loci* of genes, associated with mesenchymal GBM identity in human tumors and BTSC lines (*Figure 4F–H*). Taken all together, this evidence further indicates that *FOSL1*/FRA-1 might modulate the mesenchymal transcriptional program by regulating the chromatin accessibility of MES genes.

## *Fosl1* deletion reduces stemness and tumor growth

Ras activating mutations have been widely used to study gliomagenesis, in combination with other alterations as *Akt* mutation, loss of *Ink4a/Arf* or *Trp53* (*Friedmann-Morvinski et al., 2012*; *Holland et al., 2000*; *Koschmann et al., 2016*; *Muñoz et al., 2013*; *Uhrbom et al., 2002*). Thus, we then explored the possibility that *Fosl1* could modulate the tumorigenic potential of the p53-null *Kras* mutant cells.

Cell viability was significantly decreased in *Fosl1* KO cell lines as compared to sgCtrl (*Figure 5A*). Concomitantly, we observed a significant decreased percentage of cells in S-phase (mean values: sgCtrl = 42.6%; sgFosl1_1 = 21.6%, Student's t test p≤0.001; sgFosl1_3 = 20.4%, Student's t test p=0.003), an increase in percentage of cells in G2/M (mean values: sgCtrl = 11.7%, sgFosl1_1 = 28.4%, Student's t test p≤0.001; sgFosl1_3 = 23.4%, Student's t test p=0.012) (*Figure 5B*), and a

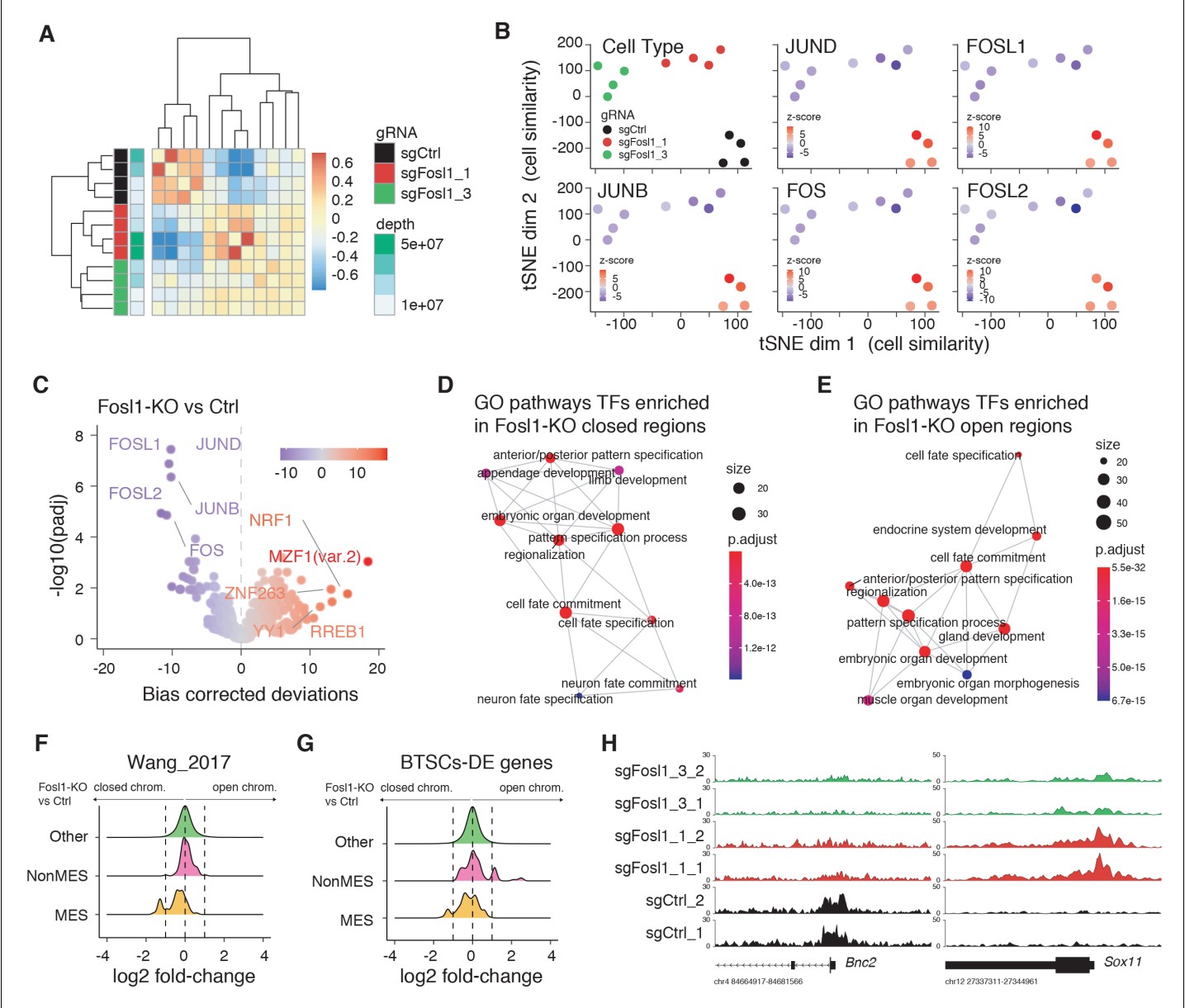

**Figure 4.** *Fosl1* depletion affects the chromatin accessibility of mesenchymal (MES) transcription program and differentiation genes in mouse glioma-initiating cells. (**A**) Correlation heatmap of the ATAC-seq samples. Clustering of the *Fosl1*-WT (sgCtrl, n = 4) and *Fosl1*-depleted (sg*Fosl1*_1 and sg*Fosl1*_3, n = 8) samples is based upon the bias corrected deviations in chromatin accessibility (see Materials and methods). (**B**) tSNE visualization of cellular similarity between *Fosl1*-depleted and control cells based on chromatin accessibility. Samples are color-coded according to the cell type (black, red, and green for sgCtrl, sgFosl1_1, and sgFosl1_3 cells, respectively), or by directional z-scores. (**C**) Volcano plot illustrating the mean difference in bias-corrected accessibility deviations between *Fosl1*-deficient and control cells against the FDR-corrected p-value for that difference. The top differential motifs are highlighted in violet and red, indicating decreased and increased accessibility, respectively. (**D**, **E**) Top enriched Gene Ontology (GO) biological processes pathways for the regions with decreased (**D**) and increased (**E**) chromatin accessibility upon *Fosl1* loss. The nodes represent the functional categories from the respective databases, color-coded by the significance of enrichment (FDR < 0.05). The node size indicates the number of query genes represented among the ontology term, and the edges highlight the relative relationships among these categories. (**F**, **G**) Density plots showing the distributions of the log2 fold-changes in chromatin accessibility of the indicated probes, as measured with *limma* by comparing *Fosl1*-KO versus control cells. (**H**) Representative ATAC-seq tracks of two technical replicates for the MES *Bnc2* and non-MES *Sox11* markers loci. Tracks are color-coded as in panels (**A**) and (**B**).

The online version of this article includes the following source data for figure 4:

**Source data 1.** Source data of *Figure 4A, C–G*.

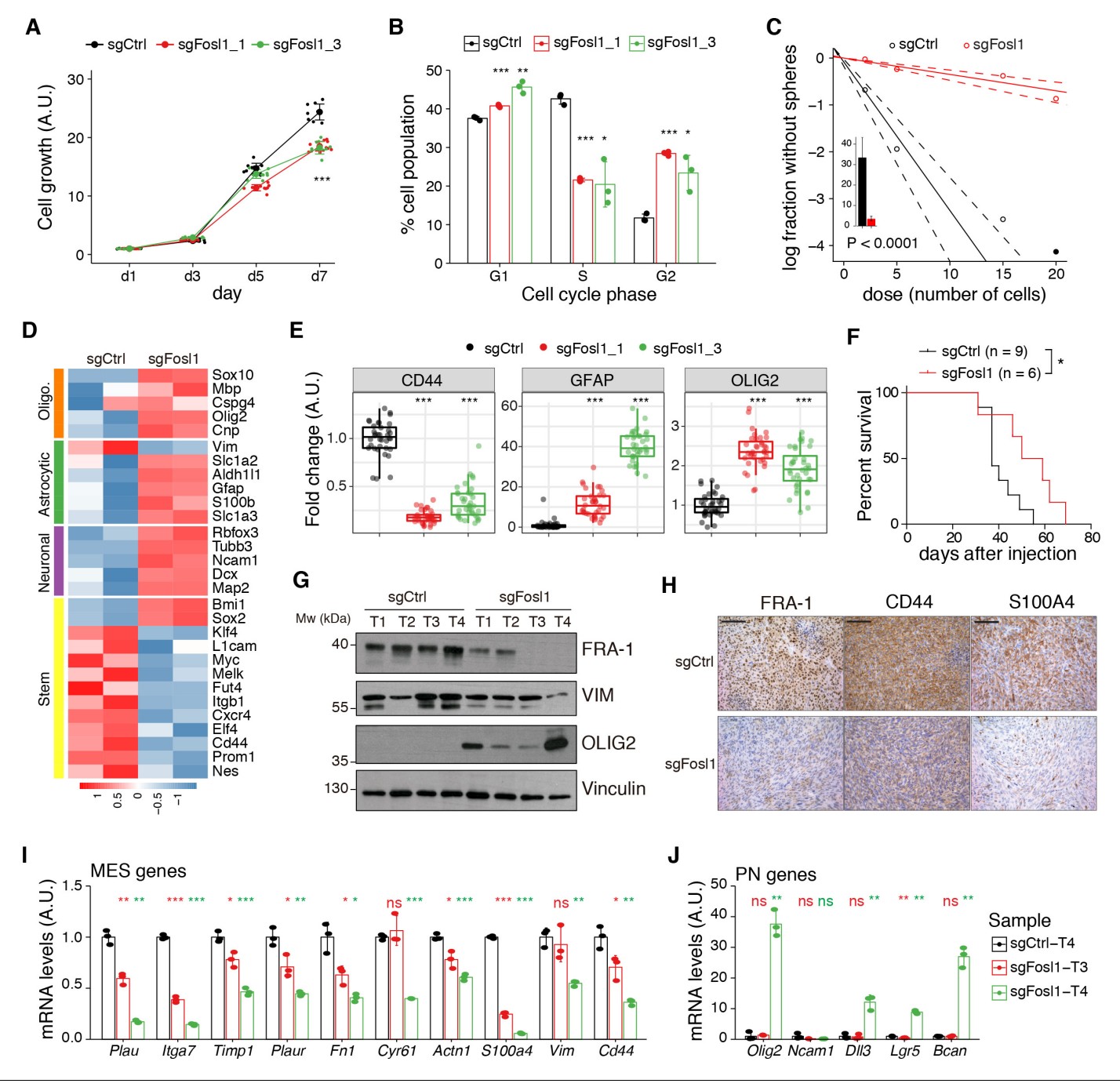

**Figure 5.** *Fosl1* knock-out (KO) impairs cell growth and stemness in vitro and increases survival in a orthotopic glioma model. (**A**) Cell viability of control and *Fosl1* KO p53-null *Kras*$^{G12V}$ neural stem cells (NSCs) measured by MTT assay; absorbance values were normalized to day 1. Data from a representative of three independent experiments are presented as mean ± SD (n = 10, technical replicates). Two-way ANOVA, relative to sgCtrl for both sg*Fosl1*_1 and sg*Fosl1*_3: ***p≤0.001. (**B**) Quantification of cell cycle populations of control and *Fosl1* KO p53-null *Kras*$^{G12V}$ NSCs by flow cytometry analysis of PI staining. Data from a representative of two independent experiments are presented as mean ± SD (n = 3, technical replicates). Student's t test, relative to sgCtrl: *p≤0.05; **p≤0.01; ***p≤0.001. (**C**) Representative limiting dilution experiment on p53-null *Kras*$^{G12V}$ sgCtrl and sg*Fosl1*_1 NSCs, calculated with extreme limiting dilution assay (ELDA) analysis; bar plot inlet shows the estimated stem cell frequency with the confidence interval; chi-square p<0.0001. (**D**) Heatmap of expression of stem cell (yellow) and lineage-specific (neuronal – purple, astrocytic – green, and oligodendrocytic – orange) genes, comparing sgCtrl and sg*Fosl1*_1 p53-null *Kras*$^{G12V}$ NSCs. Two biological replicates are shown. (**E**) Quantification of pixel area (fold-change relative to sgCtrl) of CD44, GFAP, and OLIG2 relative to DAPI pixel area per field of view in control and *Fosl1* KO p53-null *Kras*$^{G12V}$ NSCs. Data from a representative of two independent experiments; Student's t test, relative to sgCtrl: ***p≤0.001. (**F**) Kaplan–Meier survival

*Figure 5 continued on next page*

eLife Research article

Cancer Biology

curves of *nu/nu* mice injected with p53-null *Kras*^G12V^ sgCtrl (n = 9) and sg*Fosl1*_1 (n = 6) NSCs. Log-rank p=0.0263. (**G**) Western blot analysis using the indicated antibodies of four sgCtrl and four sg*Fosl1*_1 tumors (showing low or no detectable expression of FRA-1); vinculin was used as loading control. (**H**) Representative images of IHCs using the indicated antibodies. Scale bars represent 100 μm. (**I**) mRNA expression of MES genes in the samples sgCtrl–T4 (higher FRA-1 expression) and sg*Fosl1*_1–T3 and –T4 (no detectable FRA-1 expression). (**J**) mRNA expression of PN genes in samples as in (**H**). Data from a representative of two experiments are presented as mean ± SD (n = 3, technical replicates), normalized to *Gapdh* expression. Student's t test for sgFosl1_1 tumors, relative to sgCtrl–T4: ns = not significant, *p≤0.05, **p≤0.01, ***p≤0.001.

**Source data 1.** Source data of *Figure 5G*.
**Source data 2.** Source data of *Figure 5A–C, E, F, I*, J.
**Figure supplement 1.** *Fosl1* loss is associated with the reduction of proliferative genes and increase in differentiation genes.

reduction of the expression of cell cycle regulator genes (*Ccnb1*, *Ccnd1*, *Ccne1,* and *Cdk1*, among others) (*Figure 5—figure supplement 1A*).

Another aspect that contributes to GBM aggressiveness is its heterogeneity, attributable in part to the presence of GSCs. By using limiting dilution assays, we found that *Fosl1* is required for the maintenance of stem cell activity, with a stem cell frequency estimate of sgFosl1_1 = 28.6 compared to sgCtrl = 3 (chi-square p<2.2e-16) (*Figure 5C*). Moreover, RNA-seq analysis showed that sg*Fosl1*_1 cells downregulated the expression of stem genes (*Elf4*, *Klf4*, *Itgb1*, *Nes*, *Sall4*, *L1cam*, *Melk*, *Cd44*, *Myc*, *Fut4*, *Cxcr4*, *Prom1*) while upregulating the expression of lineage-specific genes: neuronal (*Map2*, *Ncam1*, *Tubb3*, *Slc1a2*, *Rbfox3*, *Dcx*), astrocytic (*Aldh1l1*, *Gfap*, *S100b*, *Slc1a3*), and oligodendrocytic (*Olig2*, *Sox10*, *Cnp*, *Mbp*, *Cspg4*) (*Figure 5D*). The different expression of some of the stem/differentiation markers was confirmed also by immunofluorescence analysis. While *Fosl1* KO cells presented low expression of the stem cell marker CD44, differentiation markers as GFAP and OLIG2 were significantly higher when compared to sgCtrl cells (*Figure 5E* and *Figure 5—figure supplement 1B*).

We then sought to test whether (i) p53-null *Kras*^G12V^ NSCs were tumorigenic and (ii) *Fosl1* played any role in their tumorigenic potential. Intracranial injections of p53-null *Kras*^G12V^ NSCs in *nu/nu* mice led to the development of high-grade tumors with a median survival of 37 days in control cells (n = 9). In contrast, sg*Fosl1*_1-injected mice (n = 6) had a significant increase in median survival (54.5 days, log-rank p=0.0263) (*Figure 5F*). Consistent with our in vitro experiments (*Figure 3D–F*), *Fosl1*-depleted tumors failed to support mesenchymal program (*Figure 5G–I*). Western blot, immunohisto-chemical, and qPCR analysis show the reduction of MES markers (VIM, CD44, and S100A4) and the expression of the PN marker OLIG2 to depend on *Fosl1* expression (*Figure 5G–J*).

Overall, our data in p53-null *Kras* mutant NSCs support the conclusion that, besides controlling cell proliferation, *Fosl1* plays a critical role in the maintenance of the stem cell activity and tumorigenicity.

## *Fosl1* amplifies mesenchymal gene expression

To further assess the role of *Fosl1* as a key player in the control of the MGS, we used a mouse model of inducible *Fosl1* overexpression containing the alleles *Kras*^LSLG12V^; *Trp53*^lox^; *ROSA26*^LSLrtTA-IRES-EGFP^; *Col1a1*^TetO-Fosl1^ (here referred to as *Fosl1*^tetON^). Similar to the loss-of-function approach here used, this allelic combination allows the expression of *Kras*^G12V^ and deletion of *p53* after Cre recombination. Moreover, the expression of the reverse tetracycline transactivator (rtTA) allows, upon induction with doxycycline (Dox), the ectopic expression of *Fosl1* (Flag tagged), under the control of the *Col1a1* locus and a tetracycline-responsive element (TRE or Tet-O) (*Belteki, 2005*; *Hasenfuss et al., 2014*).

NSCs derived from *Fosl1*^WT^ and *Fosl1*^tetON^ mice were infected in vitro with a lentiviral vector expressing the Cre recombinase and efficient infection was confirmed by fluorescence microscopy as the cells expressing the rtTA should express GFP (data not shown). FRA-1 overexpression, as well as Flag-tag expression, was then tested by western blot after 72 hr of Dox induction (*Figure 6A*). When *Fosl1*^tetON^ NSCs were analyzed by qRT-PCR for the expression of MES/PN markers, a significant upregulation of most MES genes and downregulation of PN genes was found in the cells overex-pressing *Fosl1* (*Figure 6B, C*), thereby complementing our findings in *Fosl1* KO cells.

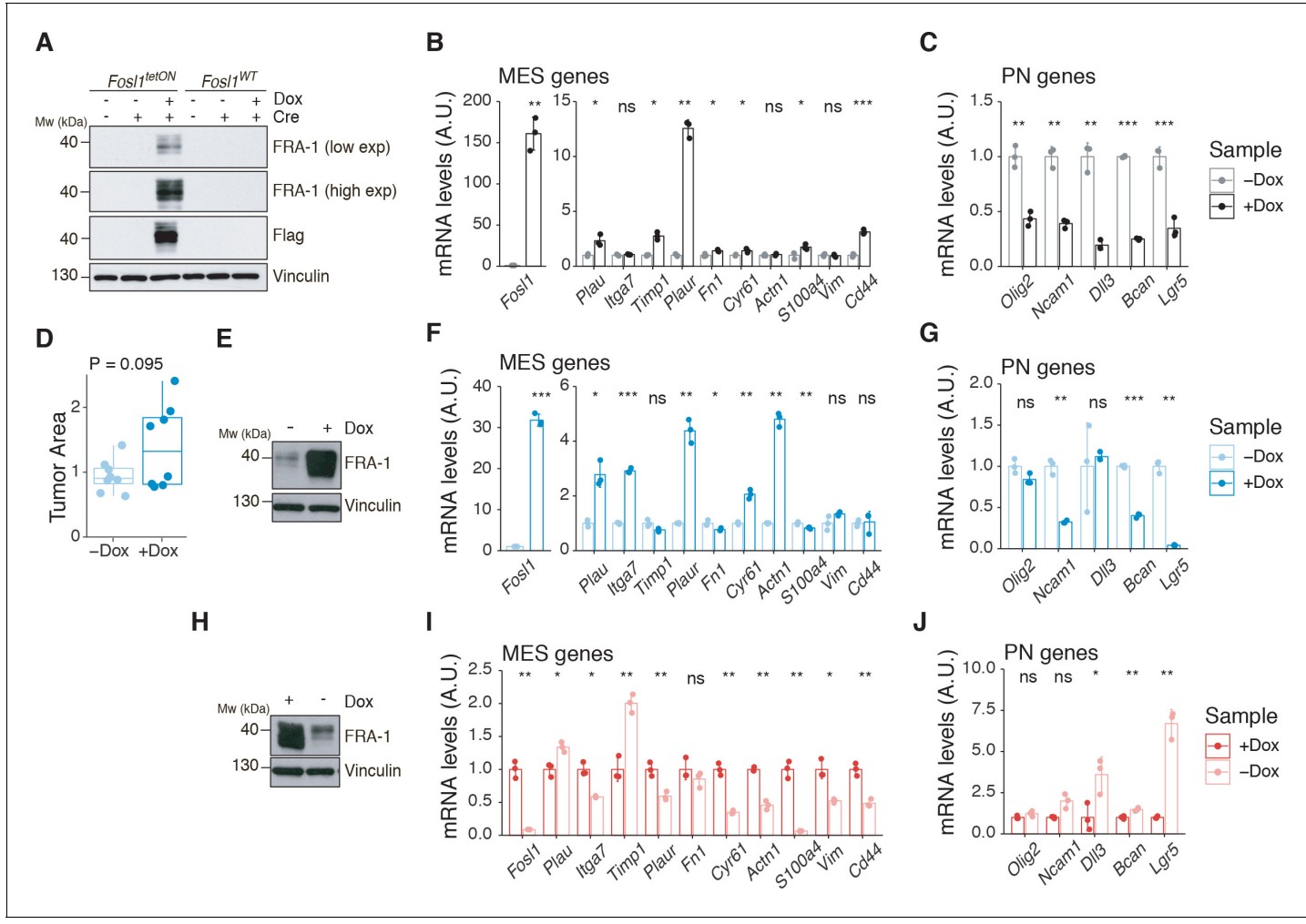

**Figure 6.** *Fosl1* overexpression upregulates the MES gene signature (MGS) and induces larger tumors in vivo. (A) Western blot analysis of FRA-1 and Flag expression on *Fosl1^tetON^* and *Fosl1^WT^* neural stem cells (NSCs) derived from *Kras^LSLG12V^*; *Trp53^lox^*; *ROSA26^LSLrtTA-IRES-EGFP^*; *Col1a1^TetO-Fosl1^* mice upon in vitro infection with Cre and induction of *Fosl1* overexpression with 1 μg/mL doxycycline (Dox) for 72 hr; vinculin was used as loading control. (B) mRNA expression of *Fosl1* and mesenchymal (MES) genes in *Fosl1^tetON^* p53-null *Kras^G12V^* cells upon 72 hr induction with 1 μg/mL Dox. (C) mRNA expression of PN genes in *Fosl1^tetON^* p53-null *Kras^G12V^* cells upon 72 hr induction with 1 μg/mL Dox. (D) Quantification of tumor area (μm²) of −Dox and +Dox tumors (n = 8/8). For each mouse, the brain section on the hematoxylin and eosin (H&E) slide with a larger tumor was considered and quantified using the ZEN software (Zeiss). (E) Western blot detection of FRA-1 expression in tumorspheres derived from a control (−Dox) tumor. Tumorspheres were isolated and kept without Dox until first passage, when 1 μg/mL Dox was added and kept for 19 days (+Dox in vitro). (F) mRNA expression of *Fosl1* and MES genes in tumorspheres in the absence or presence of Dox for 19 days. (G) mRNA expression of PN genes in tumorspheres in the absence or presence of Dox for 19 days. (H) Western blot detection of FRA-1 expression in tumorspheres derived from a *Fosl1* overexpressing (+Dox) tumor. Tumorspheres were isolated and kept with 1 μg/mL Dox until first passage, when Dox was removed for 19 days (−Dox in vitro). (I) mRNA expression of *Fosl1* and MES genes in tumorspheres in the presence or absence of Dox for 19 days. (J) mRNA expression of PN genes in tumorspheres in the presence or absence of Dox for 19 days. qRT-PCR data from a representative of two experiments are presented as mean ± SD (n = 3, technical replicates), normalized to *Gapdh* expression. Student's t test, relative to the respective control (−Dox in B, C, F, and G; +Dox in I and J): ns = not significant, *p≤0.05, **p≤0.01, ***p≤0.001.

The online version of this article includes the following source data and figure supplement(s) for figure 6:

**Source data 1.** Source data of *Figure 6A*.
**Source data 2.** Source data of *Figure 6E*.
**Source data 3.** Source data of *Figure 6H*.
**Source data 4.** Source data of *Figure 6B–D, F, G, I,*J.
**Figure supplement 1.** Characterization of *Fosl1* overexpressing mouse tumors.
**Figure supplement 1—source data 1.** Source data of *Figure 6—figure supplement 1A*.

To investigate if the MES phenotype induced with *Fosl1* overexpression would have any effect in vivo, p53-null *Kras*$^{G12V}$ *Fosl1*$^{tetON}$ NSCs were intracranially injected into syngeneic C57BL/6J wild-type mice. Injected mice were randomized and subjected to Dox diet (food pellets and drinking water) or kept as controls with regular food and drinking water with 1% sucrose. *Fosl1* overexpressing mice (+Dox) developed larger tumors that were more infiltrative and aggressive than controls (−Dox), which mostly grew as superficial tumor masses instead (*Figure 6D*). This phenotype appears to be independent of tumor cells proliferation as gauged by Ki-67 staining and does not affect overall survival (*Figure 6—figure supplement 1A, B*).

Tumorspheres were derived from −Dox and +Dox tumor-bearing mice, and *Fosl1* expression was manipulated in vitro through addition or withdrawal of Dox from the culture medium. In the case of tumorspheres derived from a −Dox tumor, when Dox was added for 19 days, high levels of FRA-1 expression were detected by western blot (*Figure 6E*). At the mRNA level, Dox treatment also greatly increased *Fosl1* expression, as well as some of the MES genes (*Figure 6F*), while the expression of PN genes was downregulated (*Figure 6G*). Conversely, when Dox was removed from +Dox-derived tumorspheres for 19 days, the expression of FRA-1 decreased (*Figure 6H, I*), along with the expression of MES genes (*Figure 6I*), while PN genes were upregulated (*Figure 6J*). These results confirm the essential role of *Fosl1* in the regulation of the MGS in p53-null *Kras*$^{G12V}$ tumor cells and the plasticity between the PN and MES subtypes.

## *FOSL1* controls growth, stemness, and mesenchymal gene expression in patient-derived BTSCs

To prove the relevance of our findings in the context of human tumors, we analyzed BTSC lines characterized as non-MES (h676, h543, and BTSC 268) or MES (BTSC 349, BTSC 380, and BTSC 233) (this study and *Ozawa et al., 2014*). By western blot, we found that consistent with what was observed either in human BTSCs (*Figure 1D*) or mouse NSCs (*Figure 3A*), MES cell lines expressed high levels of FRA-1 and activation of the MEK/ERK pathway (*Figure 7A*).

To study the role of *FOSL1* in the context of human BTSCs, its expression was modulated in the MES BTSC 380 using two Dox-inducible shRNAs (sh*FOSL1*_3 and sh*FOSL1*_10). We confirmed by western blot FRA-1 downregulation after 3 and 7 days of Dox treatment (*Figure 7B*). In line to what was observed in mouse glioma-initiating cells, *FOSL1* silencing in MES BTSC 380 resulted in reduced cell growth (*Figure 7C*) with a significant reduction of the percentage of BrdU positive cells compared to Dox-untreated cells (*Figure 7D*). Moreover, *FOSL1* silencing decreased the sphere-forming capacity of MES BTSC 380 with an estimated stem cell frequency of shGFP −Dox = 3.5, shGFP +Dox = 3.4, chi-square p=0.8457; sh*FOSL1*_3 −Dox = 4.3, sh*FOSL1*_3 +Dox = 7.6, chi-square p=0.0002195; sh*FOSL1*_10 −Dox = 5.4, sh*FOSL1*_10 +Dox = 11.1, chi-square p=5.918e-06 (*Figure 7E*). Comparable results were also obtained in the MES BTSC 349 cells (*Figure 7—figure supplement 1A–D*). In line with our mouse experiments, *FOSL1* silencing resulted in the significant downregulation of the MES genes (*Figure 7—figure supplement 1E*, *left panel*), whereas proneural gene expression was unchanged (*Figure 7—figure supplement 1E*, *right panel*). Of note, *FOSL1* silencing affected BTSCs fitness also when propagated in differentiation conditions (*Figure 7—figure supplement 1F, G*).

Similar to what was observed in mouse tumors (*Figure 6—figure supplement 1B*), *FOSL1* overexpression in two non-MES lines (h543 and h676) did not lead to changes in their proliferation capacity (*Figure 7—figure supplement 1H, I*). Most importantly, *FOSL1* silencing in these non-MES lines had no impact on cell growth (*Figure 7—figure supplement 1J, K*), underscoring a mesenchymal context-dependent role for *FOSL1* in glioma cells.

We then tested whether *FOSL1*/FRA-1 modulates the MGS via direct target regulation. To this end, we first identified high-confidence *FOSL1*/FRA-1 binding sites in chromatin immunoprecipitation-seq (ChIP-seq) previously generated in the KRAS mutant HCT116 colorectal cancer cell line (see Materials and methods), and then we determined the counts per million reads (CPM) of the enhancer histone mark H3K27Ac in a set of MES (n = 10) and non-MES BTSCs (n = 10) (*Mack et al., 2019*), selected based on the highest and lowest *FOSL1* expression, respectively. PCA showed a marked separation of the two groups of BTSCs (*Figure 7F*). Differential enrichment analysis by DESeq2 revealed 11748 regions statistically significant (FDR < 0.005) for H3K27Ac at *FOSL1*/FRA-1 binding sites in either MES or non-MES BTSCs (*Figure 7G*). Next, we compared H3K27Ac distribution over *FOSL1*/FRA-1 binding sites to that of the non-MES MR *OLIG2*. This analysis showed that

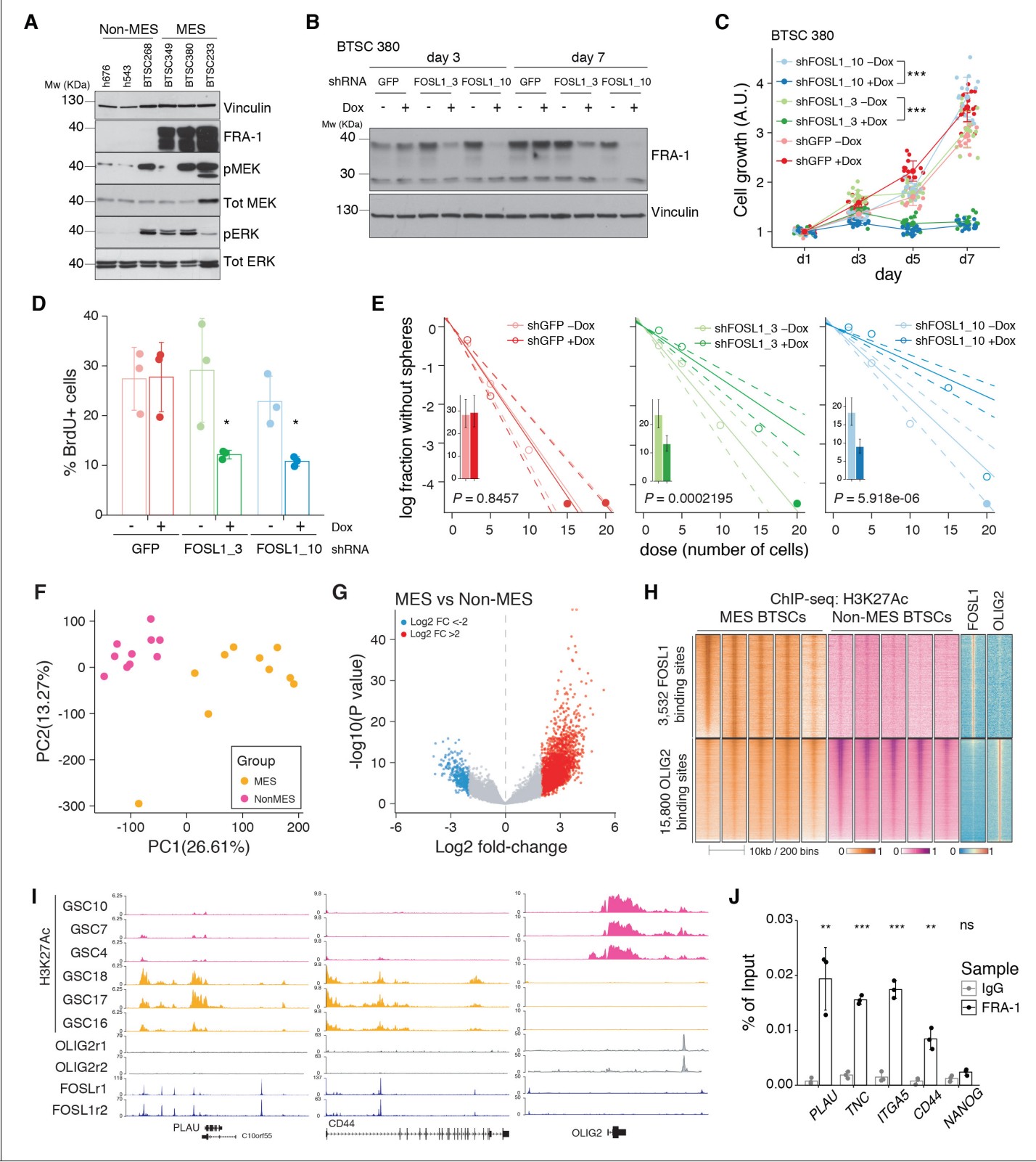

**Figure 7.** *FOSL1* contributes to mesenchymal (MES) genes activation, cell growth, and stemness in MES brain tumor stem cells (BTSCs). (**A**) Western blot analysis using the specified antibodies of human BTSC lines, characterized as non-MES (*left*) and MES (*right*). (**B**) Western blot detection of FRA-1 in MES BTSC 380 upon transduction with inducible shRNAs targeting GFP (control) and *FOSL1*, analyzed after 3 and 7 days of doxycycline (Dox) treatment; vinculin was used as loading control. (**C**) Cell growth of BTSC 380 shGFP and sh*FOSL1*, in the absence or presence of Dox, measured by

*Figure 7 continued on next page*

*Figure 7 continued*

MTT assay; absorbance values were normalized to day 1. Data from a representative of three independent experiments are presented as mean ± SD (n = 15, technical replicates). Two-way ANOVA, –Dox vs. +Dox: ***$p \leq 0.001$. (**D**) BrdU of BTSC 380 shGFP and sh*FOSL1*, in the absence or presence of Dox, analyzed by flow cytometry. Data from a representative of two independent experiments are presented as mean ± SD (n = 3, technical replicates). Student's t test, relative to the respective control (–Dox): *$p \leq 0.05$. (**E**) Representative limiting dilution analysis on BTSC380 for shGFP and sh*FOSL1*, in the presence or absence of Dox, calculated with extreme limiting dilution assay (ELDA) analysis; bar plot inlets show the estimated stem cell frequency with the confidence interval; chi-square p-values are indicated. (**F**) Principal component analysis of H3K27Ac signal over *FOSL1*/FRA-1 binding sites, calculated using MACS on ENCODE samples (see Materials and methods), in non-MES (n = 10) and MES BTSC (n = 10) (from *Mack et al., 2019*). (**G**) Volcano plot illustrating the log2 fold-change differences in H3K27Ac signal between non-MES and MES BTSCs against the p-value for that difference. Blue and red probes represent statistically significant differences (FDR < 0.005) in H3K27Ac signal between non-MES and MES BTSCs. (**H**) Heatmap of ChIP-seq enrichment of *FOSL1*/FRA-1 or *OLIG2* binding sites for the indicated profiles. (**I**) View of the *PLAU*, *CD44,* and *OLIG2* loci of selected profiles. (**J**) Representative ChIP experiment in BTSC 349 cells. The panel shows FRA-1 binding to the promoter of a subset of MES targets (n = 3, technical replicates) expressed as a percentage of the initial DNA amount in the immune-precipitated fraction. *NANOG* gene was used as a negative control. Student's t test, relative to IgG: ns = not significant, **$p \leq 0.01$, ***$p \leq 0.001$.

The online version of this article includes the following source data and figure supplement(s) for figure 7:

**Source data 1.** Source data of *Figure 7A*.
**Source data 2.** Source data of *Figure 7B*.
**Source data 3.** Source data of *Figure 7C–G*, J.
**Figure supplement 1.** Further characterization of *FOSL1* role in human brain tumor stem cells (BTSCs).
**Figure supplement 1—source data 1.** Source data of *Figure 7—figure supplement 1A*.
**Figure supplement 1—source data 2.** Source data of *Figure 7—figure supplement 1H*.
**Figure supplement 1—source data 3.** Source data of *Figure 7—figure supplement 1J*.
**Figure supplement 1—source data 4.** Source data of *Figure 7—figure supplement 1B–E, G, I, K*.

*FOSL1*/FRA-1 binding sites were systematically decorated with H3K27Ac in MES BTSCs, while the inverse trend was observed at *OLIG2* binding sites (*Figure 7H, I*). Validation by ChIP-qPCR in an independent MES BTSC line (BTSC 349) confirmed FRA-1 direct binding at promoters of some MES genes including *PLAU*, *TNC*, *ITGA5,* and *CD44* in GBM cells (*Figure 7J*).

Altogether, our data support that *FOSL1*/FRA-1 regulates MES gene expression and aggressiveness in human gliomas via direct transcriptional regulation, downstream of the NF1-MAPK-FOSL1 signaling.

## Discussion

The most broadly accepted transcriptional classification of GBM was originally based on gene expression profiles of bulk tumors (*Verhaak et al., 2010*), which did not discriminate the contribution of tumor cells and TME to the transcriptional signatures. It is now becoming evident that both cell-intrinsic and -extrinsic cues can contribute to the specification of the MES subtype (*Bhat et al., 2013*; *Hara et al., 2021*; *Neftel et al., 2019*; *Schmitt et al., 2021*; *Wang et al., 2017*). Bhat and colleagues had shown that while some of the MES GBMs maintained the mesenchymal characteristics when expanded in vitro as BTSCs, some others lost the MGS after few passages while exhibiting a higher non-MGSs (*Bhat et al., 2013*). These data, together with the evidence that xenografts into immunocompromised mice of BTSCs derived from MES GBMs were also unable to fully restore the MES phenotype (*Bhat et al., 2013*), suggested that the presence of an intact TME potentially contributed to the maintenance of a MGS. In support of this, Schmitt and colleagues have recently shown that innate immune cells divert GBM cells to a proneural-to-mesenchymal transition (PN-to-MES) that also contributes to therapeutic resistance (*Schmitt et al., 2021*).

The transcriptional GBM subtypes were lately redefined based on the expression of glioma-intrinsic genes, thus excluding the genes expressed by cells of the TME (*Richards et al., 2021*; *Wang et al., 2017*). Our MRA on the BTSCs points to the AP-1 family member *FOSL1* as one pioneer TF contributing to the cell-intrinsic MGS. Previous tumor bulk analysis identified a related AP-1 family member *FOSL2*, together with *CEBPB*, *STAT3*, and *TAZ*, as important regulators of the MES GBM subtype (*Bhat et al., 2011*; *Carro et al., 2010*). While *FOSL1* was also listed as a putative MES MR (*Carro et al., 2010*), its function and mechanism of action have not been further characterized since then. Our experimental data show that *FOSL1* is a key regulator of GBM subtype plasticity and MES transition, and define the molecular mechanism through which *FOSL1* is regulated. While here

we have focused on the TFs contributing to MES specifications, previous studies had highlighted the role of other TFs, some of which were also identified in our MRA, such as *OLIG2, SALL2,* and *ASCL1*, as important molecules for non-MES GBM cells (*Suvà et al., 2014*). Moreover, using a similar MRA, Wu and colleagues have recently described also *SOX10* as another TF that contributes to the identity of non-MES GBM cells. Strikingly, loss of *SOX10* resulted in MES transition associated with changes in chromatin accessibility in regions that are specifically enriched for FRA-1 binding motifs (*Wu et al., 2020*). Lastly, using an unbiased CRISPR/Cas9 genome-wide screening, Richards and colleagues had shown that few of the top TFs identified here, such as *FOSL1, OLIG2,* and *ASCL1*, are genes essential specifically either for MES GSCs (*FOSL1*) or for non-MES GSCs (*OLIG2* and *ASCL1*) (*Richards et al., 2021*). This evidence further strengthen the relevance of the MRA that we have performed in the identification of important regulators of GBM subtype-specific cell biology.

Although consistently defined, GBM subtypes do not represent static entities. The plasticity between subtypes happens at several levels. Besides the referred MES-to-PN change in cultured GSCs compared to the parental tumor (*Bhat et al., 2013*), a PN-to-MES shift often occurs upon treatment and recurrence. Several independent studies comparing matched pairs of primary and recurrent tumors demonstrated a tendency to shift towards a MES phenotype, associated with a worse patient survival, likely as a result of treatment-induced changes in the tumor and/or the micro-environment (*Phillips et al., 2006*; *Varn et al., 2021*; *Wang et al., 2016*; *Wang et al., 2017*). More-over, distinct subtypes/cellular states can coexist within the same tumor (*Neftel et al., 2019*; *Patel et al., 2014*; *Richards et al., 2021*; *Sottoriva et al., 2013*; *Varn et al., 2021*; *Wang et al., 2019*) and targeting these multiple cellular components could result in more effective treatments (*Wang et al., 2019*).

PN-to-MES transition is often considered an EMT-like phenomenon, associated with tumor progression (*Fedele et al., 2019*). The role of *FOSL1* in EMT has been studied in other tumor types. In breast cancer cells, *FOSL1* expression correlates with mesenchymal features and drives cancer stem cells (*Tam et al., 2013*) and the regulation of EMT seems to happen through the direct binding of FRA-1 to promoters of EMT genes such as *Tgfb1*, *Zeb1,* and *Zeb2* (*Bakiri et al., 2015*). In colorectal cancer cells, *FOSL1* was also shown to promote cancer aggressiveness through EMT by direct transcription regulation of EMT-related genes (*Diesch et al., 2014*; *Liu et al., 2015*).

It is well established that *NF1* inactivation is a major genetic event associated with the MES subtype (*Verhaak et al., 2010*; *Wang et al., 2017*). However, this is probably a late event in MES gliomagenesis as all tumors possibly arise from a PN precursor and just later in disease progression acquire *NF1* alterations that are directly associated with a transition to a MES subtype (*Ozawa et al., 2014*). Moreover, *NF1* deficiency has been linked to macrophage/microglia infiltration in the MES subtype (*Wang et al., 2017*). The fact that the enriched macrophage/microglia microenvironment is also able to modulate a MES phenotype suggests that there might be a two-way interaction between tumor cells and TME. The mechanisms of *NF1*-regulated chemotaxis and whether this relationship between the TME and MGS in GBM is causal remain elusive.

Here, we provide evidence that manipulation of *NF1* expression levels in patient-derived BTSCs has a direct consequence on the tumor-intrinsic MGS activation and that such activation can at least in part be mediated by the modulation of *FOSL1*. Among the previously validated MRs, only *CEBPB* appears also to be finely tuned by *NF1* inactivation. This suggests that among the TFs previously characterized (such as *FOSL2, STAT3, BHLHB2,* and *RUNX1*), *FOSL1* and *CEBPB* might play a dominant role in the *NF1*-mediated MES transition that occurs in a glioma cell-intrinsic manner. However, whether *FOSL1* contributes also to the cross-talk between the TME and the cell-intrinsic MGS still has to be established.

Furthermore, we show that *FOSL1* is a crucial player in glioma pathogenesis, particularly in a MAPK-driven MES GBM context (*Figure 8*). Most likely, the existence of a NF1-MAPK-FOSL1 axis goes beyond GBM pathogenesis since *FOSL1* appears to be upregulated in concomitance with *NF1* mutations in multiple tumor types (*Figure 8—figure supplement 1*). Our findings broaden its previously described role in KRAS-driven epithelial tumors, such as lung and pancreatic ductal adenocarcinoma (*Vallejo et al., 2017*). *NF1* inactivation results in Ras activation, which stimulates downstream pathways as MAPK and PI3K/Akt/mTOR. RAS/MEK/ERK signaling in turn regulates FRA-1 protein stability (*Casalino et al., 2003*; *Verde et al., 2007*). *FOSL1* mRNA expression is then most likely induced by binding of the SRF/Elk1 complex to the serum-responsive element (SRE) on *FOSL1* promoter (*Esnault et al., 2017*). At the same time, FRA-1 can then directly bind to its own promoter to

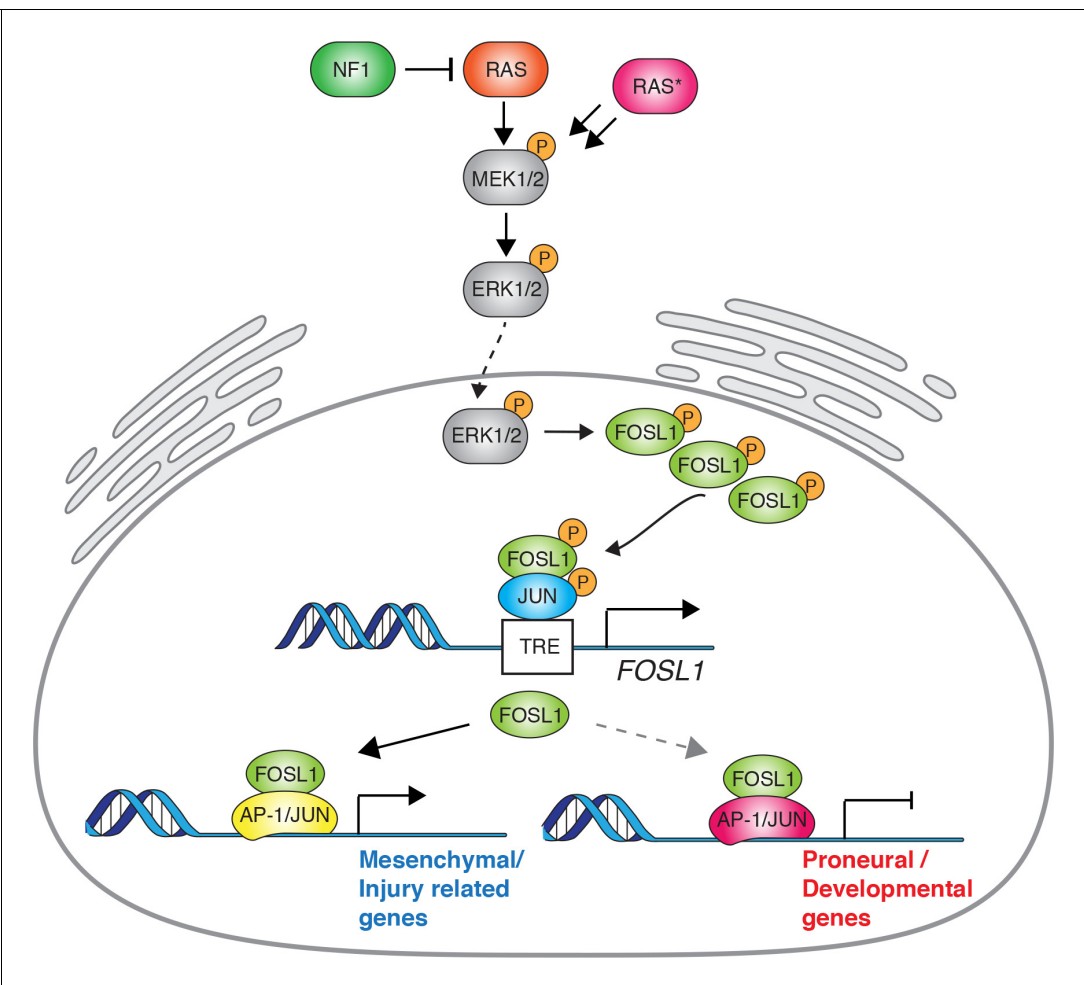

**Figure 8.** Schematic model of NF1-MAPK-FOSL1 axis in mesenchymal (MES) gliomas. NF1 alterations or RAS mutations lead to the activation of the MAPK signaling that in turn increases FOSL1 expression both at the mRNA and protein levels. FOSL1 then activates the expression of the MES gene signature and possibly inhibits the non-MES gene signature. The scheme integrates data presented in this work as well as previously published literature on the regulation of *FOSL1* expression by MAPK activation.

The online version of this article includes the following figure supplement(s) for figure 8:

**Figure supplement 1.** NF1 mutations are associated with higher *FOSL1* expression in multiple cancer types.

activate its own expression (*Diesch et al., 2014*; *Lau et al., 2016*) and those of MES genes. This further generates a feedback loop that induces MGS, increases proliferation and stemness, sustaining tumor growth. FRA-1 requires, for its transcriptional activity, heterodimerization with the AP-1 TFs JUN, JUNB, or JUND (*Eferl and Wagner, 2003*). Which of the JUN family members participate in the MES gene regulation and whether *FOSL1*/FRA-1 activates MES gene expression and simultaneously represses non-MES genes requires further investigation. Of note, pancancer analysis of anatomically distinct solid tumors suggested that c-JUN/JUNB and FOSL1/2 are bona fide canonical AP-1 TF configurations in mesenchymal states of lung, kidney, and stomach cancers (*Serresi et al., 2021*). Intriguingly, in support of a direct role in the repression of non-MES genes in GBM cells, it has been hypothesized, though not formally demonstrated, that *FOSL1*/FRA-1 could act as a transcriptional repressor of a core set of neurodevelopmental TFs, including *OLIG2* and *SALL2* (*Fiscon et al., 2018*).

In conclusion, *FOSL1*/FRA-1 is a key regulator of the MES subtype of GBM, significantly contributing to its stem cell features, which could open new therapeutic options. Although *FOSL1*/FRA-1 pharmacological inhibition is difficult to achieve due to its enzymatic activity, a gene therapy

approach targeting *FOSL1*/FRA-1 expression through CRISPR/Cas9 or PROTAC, for instance, could constitute attractive alternatives to treat mesenchymal GBM patients.

# Materials and methods

## Key resources table

| Reagent type (species) or resource | Designation | Source or reference | Identifiers | Additional information |
|---|---|---|---|---|
| Antibody | Anti-FRA-1 (Rabbit polyclonal) | Santa Cruz Biotechnology | Cat#sc-183; RRID:AB_2106928 | WB(1:1000) |
| Antibody | Anti-FRA-1 (Rabbit polyclonal) | Santa Cruz Biotechnology | Cat#sc-605, RRID:AB_2106927 | WB(1:1000) |
| Antibody | anti-CD44 (Rat monoclonal) | BD Biosciences | Cat#550538; RRID:AB_393732 | IF(1:100) |
| Antibody | Anti-S100A4 (Rabbit polyclonal) | Abcam | Cat#ab27957, RRID:AB_2183775 | IHC(1:300) |
| Antibody | Anti-Ki67 (Rabbit monoclonal) | Master Diagnostica | Cat#000310QD | IHC(undiluted) |
| Antibody | Anti-FLAG (DYKDDDDK Tag) (Rabbit polyclonal) | Cell Signaling Technology | Cat#2368, RRID:AB_2217020 | WB(1:2000) |
| Antibody | Anti-GFAP (Mouse monoclonal) | Sigma-Aldrich | Cat#G3893, RRID:AB_477010 | WB(1:5000) |
| Antibody | Anti-GFAP (Mouse monoclonal) | Millipore | Cat#MAB360, RRID:AB_11212597 | IF(1:400) |
| Antibody | Anti-NF1 (Rabbit polyclonal) | Santa Cruz Biotechnology | Cat#sc-67, RRID:AB_2149681 | WB(1:500) |
| Antibody | Anti-NF1 (Rabbit polyclonal) | Bethyl | Cat#A300-140A, RRID:AB_2149790 | WB(1:1000) |
| Antibody | Anti-OLIG2 (Rabbit polyclonal) | Millipore | Cat#AB9610, RRID:AB_570666 | WB(1:2000) |
| Antibody | Anti-VIMENTIN (Rabbit monoclonal) | Cell Signaling Technology | Cat#5741, RRID:AB_10695459 | WB(1:3000) |
| Antibody | Anti-phospho-p44/42 MAPK (Erk1/2) (Thr202/Tyr204) (Rabbit polyclonal) | Cell Signaling Technology | Cat#9101, RRID:AB_331646 | WB(1:2000) |
| Antibody | Anti-p44/42 MAPK (Erk1/2) (Rabbit polyclonal) | Cell Signaling Technology | Cat#9102, RRID:AB_330744 | WB(1:1000) |
| Antibody | Anti-phospho-MEK1/2 (Ser217/221) (Rabbit polyclonal) | Cell Signaling Technology | Cat#9154, RRID:AB_2138017 | WB(1:500) |
| Antibody | Anti-MEK1/2 (Rabbit polyclonal) | Cell Signaling Technology | Cat#9122, RRID:AB_823567 | WB(1:1000) |
| Antibody | Anti-human YKL40 (Rabbit polyclonal) | Qidel | Cat#4815, RRID:AB_452475 | WB(1:1000) |
| Antibody | Anti-PI3 kinase, p85 (Rabbit polyclonal) | Millipore | Cat#06-195, RRID:AB_310069 | WB(1:10,000) |
| Antibody | Anti-vinculin (Mouse monoclonal) | Sigma-Aldrich | Cat#V9131, RRID:AB_477629 | WB(1:10,000) |
| Antibody | Anti-α-tubulin (Mouse monoclonal) | Abcam | Cat#ab7291, RRID:AB_2241126 | WB(1:10,000) |

*Continued on next page*

Continued

| Reagent type (species) or resource | Designation | Source or reference | Identifiers | Additional information |
|---|---|---|---|---|
| Antibody | Biotinylated anti-rabbit IgG (Goat polyclonal) | Vector Laboratories | Cat#BA-1000, RRID:AB_2313606 | IHC(1:200) |
| Antibody | Anti-rat IgG (H+L) (goat unknown) | Vector Laboratories | Cat#BA-9400, RRID:AB_2336202 | IHC(1:200) |
| Antibody | Peroxidase-AffiniPure anti-mouse IgG (Goat polyclonal) | Jackson Immuno Research Labs | Cat#115-035-003, RRID:AB_10015289 | WB(1:10,000) |
| Antibody | Peroxidase-AffiniPure anti-rabbit IgG (Goat polyclonal) | Jackson Immuno Research Labs | Cat#111-035-003, RRID:AB_2313567 | WB(1:10,000) |
| Antibody | Alexa Fluor 488 anti-rabbit IgG (H+L) (Donkey polyclonal) | Thermo Fisher Scientific | Cat#A21206; RRID:AB_2535792 | IF(1:400) |
| Antibody | Alexa Fluor 488 anti-mouse IgG (H+L) (Donkey polyclonal) | Thermo Fisher Scientific | Cat#A21202; RRID:AB_141607 | IF(1:400) |
| Antibody | Alexa Fluor 594 anti-rat IgG (H+L) (Donkey polyclonal) | Thermo Fisher Scientific | Cat#A21209; RRID:AB_2535795 | IF(1:400) |
| Chemical compound, drug | Ovomucoid | Worthington | Cat#LS003087 | |
| Chemical compound, drug | N-acetyl-L-cysteine | Sigma-Aldrich | Cat#A9165 | |
| Peptide, recombinant protein | Recombinant human EGF | Gibco | Cat#PHG0313 | |
| Peptide, recombinant protein | Basic-FGF | Millipore | Cat#GF003-AF | |
| Peptide, recombinant protein | Heparin | Stem Cell Technologies | Cat#07980 | |
| Chemical compound, drug | L-glutamine | Hyclone | Cat#SH3003401 | |
| Chemical compound, drug | Accumax | Thermo Fisher Scientific | Cat#00-4666-56 | |
| Chemical compound, drug | Polybrene | Sigma-Aldrich | Cat#H9268 | |
| Chemical compound, drug | Puromycin | Sigma-Aldrich | Cat#P8833 | |
| Chemical compound, drug | Doxycycline | PanReac AppliChem | Cat#A29510025 | |

*Continued*

| Reagent type (species) or resource | Designation | Source or reference | Identifiers | Additional information |
|---|---|---|---|---|
| Chemical compound, drug | Hydrogen peroxide | Sigma-Aldrich | Cat#H1009 | |
| Peptide, recombinant protein | BSA | Sigma-Aldrich | Cat#A7906 | |
| Chemical compound, drug | BrdU | Sigma-Aldrich | Cat#B9285 | |
| Chemical compound, drug | Peroxidase substrate DAB | Vector Laboratories | Cat#SK-4100 | |
| Chemical compound, drug | TRIzol | Invitrogen | Cat#15596-026 | |
| Chemical compound, drug | MTT | Sigma-Aldrich | Cat#M5655 | |
| Chemical compound, drug | PI | Sigma-Aldrich | Cat#P4170 | |
| Other | Goat serum | Sigma-Aldrich | Cat#G9023 | |
| Other | RNase A | Roche | Cat#10109142001 | |
| Other | DAPI | Sigma-Aldrich | Cat#D8417 | |
| Other | ProLong Gold Antifade | Invitrogen | Cat#P10144 | |
| Other | Protein A/G plus-agarose beads | Santa Cruz Biotechnology | Cat#sc-2003 | |
| Other | Salmon sperm DNA | Thermo Fisher Scientific | Cat#AM9680 | |
| Other | Neurobasal medium | Gibco | Cat#10888022 | |
| Other | B27 supplement | Gibco | Cat#12587010 | |
| Other | N2 supplement | Gibco | Cat#17502048 | |
| Other | Earl's Balanced Salt Solution | Gibco | Cat#14155-08 | |
| Other | Papain | Worthington | Cat#LS003119 | |
| Other | DNaseI | Roche | Cat#10104159001 | |
| Other | Mouse NeuroCult basal medium | Stem Cell Technologies | Cat#05700 | |
| Other | Mouse NeuroCult Proliferation supplement | Stem Cell Technologies | Cat#05701 | |
| Other | ACK lysing buffer | Gibco | Cat#A1049201 | |
| Other | DMEM | Sigma-Aldrich | Cat#D5796 | |
| Commercial assay or kit | High Capacity cDNA Reverse Transcription Kit | Applied Biosystems | Cat#4368814 | |
| Commercial assay or kit | SYBR Select Master Mix | Applied Biosystems | Cat#4472908 | |
| Commercial assay or kit | SuperscriptIII reverse transcriptase | Life Technologies | Cat#18080-085 | |

*Continued on next page*

*Continued*

| Reagent type (species) or resource | Designation | Source or reference | Identifiers | Additional information |
|---|---|---|---|---|
| Commercial assay or kit | QuantSeq 3′ mRNA-Seq Library Prep Kit (FWD) for Illumina | Lexogen | Cat#015 | |
| Commercial assay or kit | StemPro Osteogenesis Differentiation Kit | Life Technologies | Cat#A1007201 | |
| Commercial assay or kit | Active Ras pull down assay kit | Thermo Fisher Scientific | Cat#16117 | |
| Commercial assay or kit | QIAquick PCR purification kit | QIAGEN | Cat#28104 | |
| Commercial assay or kit | QIAGEN PCR cloning kit | QIAGEN | Cat#231124 | |
| Recombinant DNA reagent | pCHMWS-NF1-GRD | This paper | N/A | NF1-GRD overexpressing construct generated in the Carro's lab |
| Recombinant DNA reagent | pLKO-shNF1 | Sigma-Aldrich | TRCN0000238778 | |
| Recombinant DNA reagent | pGIPZ-shNF1 | This paper | N/A | Human NF1 shRNA construct generated in the Carro's lab |
| Recombinant DNA reagent | pGIPZ-shNF1 clone V2LHS_76027 (clone 4) | Open Biosystems | RHS4430-98894408 | |
| Recombinant DNA reagent | pGIPZ-shNF1 clone V2LHS_260806 (clone 5) | Open Biosystems | RHS4430-98912463 | |
| Recombinant DNA reagent | pKLV-U6gRNA-PGKpuro2ABFP | Kosuke Yusa (Wellcome Sanger Institute) | Addgene plasmid #50946 | |
| Recombinant DNA reagent | pLVX-Cre | Maria A. Blasco (CNIO) | N/A | |
| Recombinant DNA reagent | pLKO.1-TET-shFOSL1_3 and shFOSL1_10 | Silve Vicent (CIMA) | N/A | |
| Recombinant DNA reagent | pMD2.G | Carro's lab | Addgene plasmid #12259 | |
| Recombinant DNA reagent | psPAX2 | Carro's lab | Addgene plasmid #12260 | |
| Recombinant DNA reagent | pBabe-FOSL1 | *Matsuo et al., 2000* | N/A | |
| Recombinant DNA reagent | pSIN-EF1-puro-FLAG-FOSL1 | Silve Vicent (CIMA) | N/A | |
| Recombinant DNA reagent | pSIN-EF1-puro-eGFP | Silve Vicent (CIMA) | N/A | |
| Recombinant DNA reagent | RCAS-sgNf1 | This paper | N/A | Mouse Nf1 sgRNA construct generated in the Squatrito's lab |
| Recombinant DNA reagent | RCAS-shNf1 | *Ozawa et al., 2014* | N/A | |

*Continued on next page*

*Continued*

| Reagent type (species) or resource | Designation | Source or reference | Identifiers | Additional information |
|---|---|---|---|---|
| Recombinant DNA reagent | RCAS-KrasG12V | This paper | N/A | KRASG12V construct generated in the Squatrito's lab |
| Software, algorithm | FlowJo v10 | BD (Becton, Dickinson and Company) | N/A | |
| Software, algorithm | RStudio | https://rstudio.com/products/rstudio/ | N/A | |
| Software, algorithm | Nextpresso RNA-Seq pipeline | *Graña et al., 2018* | https://hub.docker.com/r/osvaldogc/nextpresso | |
| Software, algorithm | deepTools2 | *Ramírez et al., 2016* | https://deeptools.readthedocs.io/en/develop/ | |
| Software, algorithm | bowtie2 v2.3.5 | *Langmead and Salzberg, 2012* | Bowtie 2, RRID:SCR_016368 | |
| Software, algorithm | SeqMonk | https://www.bioinformatics.babraham.ac.uk/projects/seqmonk/ | SeqMonk, RRID:SCR_001913 | |
| Software, algorithm | ChaSE | *Younesy et al., 2016* | http://chase.cs.univie.ac.at/overview | |
| Software, algorithm | GSEA | *Subramanian et al., 2005* | Gene Set Enrichment Analysis, RRID:SCR_003199 | |
| Software, algorithm | R programming language | R Core team 2013 | R Project for Statistical Computing, RRID:SCR_001905 | |
| Software, algorithm | trim-galore v0.6.2 | https://www.bioinformatics.babraham.ac.uk/projects/trim_galore/ | N/A | |
| Cell line (*Gallus gallus*) | DF1 | ATCC | Cat#CRL-12203 | |
| Cell line (*Homo sapiens*) | Gp2-293 | Clontech | Cat#631458 | |
| Cell line (*Homo sapiens*) | BTSC 232 | *Fedele et al., 2017* | N/A | |
| Cell line (*Homo sapiens*) | BTSC 233, BTSC 3021, BTSC 3047, BTSC 349, BTSC 380 | This paper | N/A | Human patient-derived lines generated at Freiburg University; see Materials and methods for details |
| Cell line (*Homo sapiens*) | h543, h676 | *Ozawa et al., 2014* | N/A | |

## Generation of the BTSCs dataset and MRA

The BTSC lines dataset (n = 144) was assembled with new and previously generated transcriptomic data: 22 Illumina HumanHT-12v4 expression BeadChip microarrays newly generated at Freiburg University (GSE137310, this study), 44 RNA-seq samples (Illumina HiSeq 2500) from GSE119834 (*Mack et al., 2019*), 14 RNA-seq samples (Illumina HiSeq 2000) from SRP057855 (*Cusulin et al.,*

*2015*), 30 Affymetrix Human Genome U219 microarrays from GSE67089 (*Mao et al., 2013*), 17 Affymetrix Human Genome U133 Plus 2.0 microarrays from GSE8049 (*Günther et al., 2008*), and 17 Affymetrix GeneChip Human Genome U133A 2.0 microarrays from GSE49161 (*Bhat et al., 2013*). To analyze the RNA-seq samples, we used the Nextpresso pipeline (*Graña et al., 2018*). For the Affymetrix microarrays, raw data were downloaded from the GEO repository (https://www.ncbi.nlm.nih.gov/geo/) and subsequently the 'affy' R package (*Gautier et al., 2004*) was used for robust multi-array average normalization followed by quantile normalization. For genes with several probe sets, the median of all probes had been chosen and only common genes among all the datasets (n = 9889) were used for further analysis. To avoid issues with the use of different transcriptomic platforms, each dataset was then scaled (mean = 0, SD = 1) before assembling the combined final dataset. Transcriptional subtypes were obtained using the 'ssgsea.GBM.classification' R package (*Wang et al., 2017*), using 1000 permutations. For differential gene expression studies, we selected the 133 BTSCs lines that had concordant ssGSEA results, with MES BTSCs classified both as MES and INJ and non-MES BTSCs classified both as PN/CL and DEV. Differential gene expression (MES vs. non-MES BTSCs) was performed using the 'limma' R package (*Ritchie et al., 2015*), taking into account the possible batch differences due the different datasets assembled.

The MRA was performed using the 'RTN' R package (*Castro et al., 2016*). Normalized BTSC expression data were used as input to build a transcriptional network (TN) for 887 TFs present in the dataset. TF annotations were obtained from the human TF atlas version 1.0.1 (http://humantfs.ccbr.utoronto.ca/) (*Lambert et al., 2018*). p-Values for network edges were computed from a pooled null distribution using 1000 permutations. Edges with an adjusted-p-value<0.05 were kept for data processing inequality (DPI) filtering. In the TN, each target can be connected to multiple TFs and regulation can occur as a result of both direct and indirect interactions. DPI filtering removes the weakest interaction in any triangle of two TFs and a target gene, therefore preserving the dominant TF-target pairs and resulting in a filtered TN that highlights the most significant interactions (*Fletcher et al., 2013*). Post-DPI filtering, the MRA computes the overlap between the transcriptional regulatory unities (regulons) and the input signature genes using the hypergeometric distribution (with multiple hypothesis testing corrections). To identify MRs, the differential gene expression between MES and non-MES was used as a phenotype.

## TCGA and CGGA data analysis

RSEM normalized RNA-seq data for the TCGA GBMLGG and CGGA datasets were downloaded from the Broad Institute Firebrowse (http://gdac.broadinstitute.org) and the Chinese Glioma Genome Atlas (updated November 2019) (http://www.cgga.org.cn/), respectively. *NF1* copy number alterations and point mutations for the TCGA GBMLGG were obtained at the cBioPortal (https://www.cbioportal.org). *FOSL1* expression and *NF1* mutational status for the TCGA datasets in *Figure 8—figure supplement 1* were obtained from the Timer2.0 web portal (http://timer.cistrome.org/) (*Li et al., 2020*). Transcriptional subtypes were inferred using the 'ssgsea.GBM.classification' R package as indicated above. Glioma molecular subtypes information was downloaded from the GlioVis web portal (http://gliovis.bioinfo.cnio.es) (*Bowman et al., 2017*). Survival analysis was performed using the 'survival' R package. Stratification of the patients has been done by comparing the patients with the 30% *FOSL1* high expression to the 30% *FOSL1* low expression.

## scRNA-seq datasets

Preprocessed scRNA-seq data were downloaded from the Broad Institute Single-Cell Portal (https://singlecell.broadinstitute.org/single_cell/), study numbers SCP503 (*Richards et al., 2021*) and SCP393 (*Neftel et al., 2019*).

## Gene expression array and GSEA

For gene expression profiling of the BTSC lines of the Freiburg dataset, total RNA was prepared using the RNeasy kit (QIAGEN #74104) or the AllPrep DNA/RNA/Protein mini kit (QIAGEN #80004) and quantified using 2100 Bioanalyzer (Agilent). One-and-a-half microgram of total RNA for each sample was sent to the genomic facility of the German Cancer Research Center (DKFZ) in Heidelberg (Germany), where hybridization and data normalization were performed. Hybridization was carried out on Illumina HumanHT-12v4 expression BeadChip. GSEA was performed using the GSEA

software (http://www.broadinstitute.org/gsea/index.jsp) (*Subramanian et al., 2005*). Gene signatures are listed in *Supplementary file 5*.

## ATAC-seq

ATAC-seq was performed on 60,000 p53-null Kras$^{G12V}$ mouse NSCs transduced with either sgFosl1_1, sgFosl1_3, or sgCtrl. Briefly, the cells were pelleted in PBS and tagmentation was performed in either 50 µL of master mix containing 25 µL 2xTD buffer, 2.5 µL transposase, and 22.5 µL nuclease-free water (Nextera DNA Library Prep, Illumina #FC-121-1030) or in 50 µL of tagmentation mix containing 25 µL of TAPS-DMF buffer (80 mM TAPS, 40 mM MgCl$_2$, 50% vol/vol DMF), 0.625 µL in-house-produced hyperactive Tn5 enzyme and 24.4 µL nuclease-free water (adapted from *Hennig et al., 2018*). The tagmentation reactions were incubated for 1 hr at 37°C with moderate agitation (500–800 rpm). After the incubation, 5 µL of Proteinase K (Invitrogen #AM2548) were added to the samples to stop the transposition. The cells were subsequently lysed by adding 50 µL of AL buffer (QIAGEN #19075) and incubating for 10 min at 56°C. The DNA was extracted by means of 1.8× vol/vol AMPure XP beads (Beckman Coulter #A63881) and eluted in 18 µL. To determine the optimal number of PCR cycles required for library amplification, 2 µL of each sample were taken as template for qPCR using KAPA HiFi HotStart ReadyMix (Roche #7958927001) and 1xEvaGreen Dye (Biotium #31000). The whole probe amplification was performed in 50 µL qPCR volume with 8–12 µL of template DNA. Primers were previously described (*Buenrostro et al., 2013*). Each library was individually quantified utilizing the Qubit 3.0 Fluorometer (Life Technologies). The appropriate ATAC-seq libraries laddering pattern was determined with TapeStation High Sensitivity D1000 ScreenTapes (Agilent #5067-5584). The libraries were sequenced on the Illumina NextSeq500 using the High Output V2 150 cycles chemistry kit in a 2 × 75 bp mode.

## ATAC-seq analysis

The reads were adaptor-trimmed using the trim-galore v0.6.2 *-nextera* (https://www.bioinformatics.babraham.ac.uk/projects/trim_galore/). The mapping was conducted using the bowtie2 v2.3.5 (*Langmead and Salzberg, 2012*) default parameters. The differential chromatin accessibility analysis was performed in SeqMonk taking mouse CpG islands (mCGI) and TSSs ± 500 bp as the probe set (GRCm38 assembly). Counts were normalized by means of the 'Read Count Quantification' function with additional correction for total count (CPM), log transformation, and size factor normalization. The differential accessibility between the *Fosl1*-WT and *Fosl1*-depleted cells was determined utilizing the limma pipeline (*Ritchie et al., 2015*).

The 'chromVAR' R package (*Schep et al., 2017*) was used to analyze the chromatin accessibility data, perform corrections for known technical biases, and identify motifs with differential deviation in chromatin accessibility between the samples. The enrichGO function from the 'clusterProfiler' R package (*Yu et al., 2012*) was used to visualize the relevant pathways in *Figure 4D, E*. The bamCoverage function of the deepTools2 tool (*Ramírez et al., 2016*) was used to generate BigWig from aligned files for subsequent visualization with the 'karyoploteR' R package (*Gel and Serra, 2017*).

## ChIP-seq analysis

We downloaded FOSL1 ChIP-seq profiling from the KRAS mutant HCT116 cell line ENCODE tracks ENCFF000OZR and ENCFF000OZQ. FOSL1/FRA-1 peaks (29,738) were identified with SeqMonk using the MACS algorithm (*Zhang et al., 2008*) with a $10^{-7}$p-value cutoff. OLIG2 binding sites and ChIP-seq profiles were downloaded from GEO: GSM1306365_MGG8TPC.OLIG2r1c and GSM1306367_MGG8TPC.OLIG2r2 (*Suvà et al., 2014*). H3K27Ac data were downloaded from GSE119755 (*Mack et al., 2019*) for GSM3382275_GSC1_H3K27AC, GSM3382277_GSC10_H3K27AC, GSM3382285_GSC14_H3K27AC, GSM3382289_GSC16_H3K27AC, GSM3382291_GSC17_H3K27AC, GSM3382293_GSC18_H3K27AC, GSM3382295_GSC19_H3K27AC, GSM3382299_GSC20_H3K27AC, GSM3382303_GSC22_H3K27AC, GSM3382313_GSC27_H3K27AC, GSM3382319_GSC3_H3K27AC, GSM3382327_GSC33_H3K27AC, GSM3382331_GSC35_H3K27AC, GSM3382333_GSC36_H3K27AC, GSM3382341_GSC4_H3K27AC, GSM3382343_GSC40_H3K27AC, GSM3382345_GSC41_H3K27AC, GSM3382347_GSC43_H3K27AC, GSM3382351_GSC5_H3K27AC, GSM3382355_GSC7_H3K27AC. Quantitation was Read Count Quantitation using all reads correcting for total count only in probes normalized to the largest store, log transformed, and duplicates

ignored. Differential H3K27Ac signal was measured through the DESeq2 pipeline (*Love et al., 2014*). Heatmaps were generated with ChaSE (*Younesy et al., 2016*), plotting H3K27AC signal over FOSL1 (3532 probes, MES vs. non-MES log2 fold-change >2) or OLIG2 (15800 probes) peaks, with a 10,000 bp probe window.

## Mouse strains and husbandry

*GFAP-tv-a; hGFAP-Cre; Rosa26-LSL-Cas9* mice were previously described (*Oldrini et al., 2018*). *Kras^{LSLG12V}; Trp53^{lox}; Rosa26^{LSLrtTA-IRES-EGFP}; Col1a1^{TetO-Fosl1}* mouse strain corresponds to the MGI Allele References 3582830, 1931011, 3583817, and 5585716, respectively. Immunodeficient *nu/nu* mice (MGI: 1856108) were obtained at the Spanish National Cancer Research Centre Animal Facility.

Mice were housed in the specific pathogen-free animal house of the Spanish National Cancer Research Centre under conditions in accordance with the recommendations of the Federation of European Laboratory Animal Science Associations (FELASA). All animal experiments were approved by the Ethical Committee (CEIyBA) (# CBA 31_2019-V2) and performed in accordance with the guidelines stated in the International Guiding Principles for Biomedical Research Involving Animals, developed by the Council for International Organizations of Medical Sciences (CIOMS).

## Cell lines, cell culture, and drug treatments

Mouse NSCs were derived from the whole brain of newborn mice of *Gtv-a; hGFAP-Cre; LSL-Cas9; Trp53^{lox}* (referred to as p53-null NSCs) and *Kras^{LSLG12V}; Trp53^{lox}; Rosa26^{LSLrtTA-IRES-EGFP}; Col1a1^{TetO-Fosl1}* (referred to as *Fosl1^{TetON}* NSCs). Tumorsphere lines were derived from tumors of C57BL/6J injected with *Fosl1^{TetON}* NSCs when mice were sacrificed after showing symptoms of brain tumor disease. For the derivation of mouse NSCs and tumorspheres, tissue was enzymatically digested with 5 mL of papain digestion solution (0.94 mg/mL papain [Worthington #LS003119], 0.48 mM EDTA, 0.18 mg/mL N-acetyl-L-cysteine [Sigma-Aldrich #A9165] in Earl's Balanced Salt Solution [Gibco #14155-08]) and incubated at 37˚C for 8 min. After digestion, the enzyme was inactivated by the addition of 2 mL of 0.71 mg/mL ovomucoid (Worthington #LS003087) and 0.06 mg/mL DNaseI (Roche #10104159001) diluted in Mouse NeuroCult basal medium (Stem Cell Technologies #05700) without growth factors. Cell suspension was centrifuged at a low speed and then passed through a 40 µm mesh filter to remove undigested tissue, washed first with PBS and then with ACK lysing buffer (Gibco #A1049201) to remove red blood cells. NSCs and tumorspheres were grown in Mouse NeuroCult basal medium, supplemented with Proliferation supplement (Stem Cell Technologies #05701), 20 ng/mL recombinant human EGF (Gibco #PHG0313), 10 ng/mL basic-FGF (Millipore #GF003-AF), 2 µg/mL heparin (Stem Cell Technologies #07980), and L-glutamine (2 mM, Hyclone #SH3003401). Spheres were dissociated with Accumax (Thermo Fisher Scientific #00-4666-56) and re-plated every 4–5 days.

Patient-derived GBM stem cells (BTSCs) from Freiburg University were prepared from tumor specimens under IRB-approved guidelines (#407/09_120965) as described before (*Fedele et al., 2017*). The gender of the main BTSCs lines used in this study are BTSC 232 (male), BTSC 233 (female), BTSC 349 (female), and BTSC 380 (male). BTSCs were grown as neurospheres in Neurobasal medium (Gibco #10888022) containing B27 supplement (Gibco #12587010), N2 supplement (Gibco #17502048), b-FGF (20 ng/mL), EGF (20 ng/mL), LIF (10 ng/mL, CellGS #GFH200-20), and 2 µg/mL heparin and L-glutamine (2 mM). Patient-derived GBM stem cell lines h543 and h676, kindly provided by Eric C. Holland, were grown as neurospheres Human NeuroCult basal medium, supplemented with Proliferation supplement (Stem Cell Technologies #05751) and growth factors, as the mouse NSCs or tumorspheres (see above). BTSC 380 were differentiated by culturing them for 5 days with 0.5% FBS and 5 ng/mL of TNFalpha (Tebu-bio #300-01A-A) (*Schmitt et al., 2021*). All cell lines used were routinely tested for mycoplasma contamination by PCR.

The MAPK inhibitors GDC-0623, trametinib, and U0126 were purchased from Merck (Cat# AMBH303C5C40), Selleckchem (Cat# S2673), and Sigma-Aldrich (Cat# 662005), respectively.

## Vectors, virus production, and infection

Flag-tagged NF1-GRD (aminoacids 1131–1534) was amplified by PCR from human cortical tissue (epilepsy patient) and first cloned in the pDRIVE vector (QIAGEN #231124). Primers are listed in *Supplementary file 6*. The NF1-GRD sequence was then excised by restriction digestion using PmeI

and SpeI enzymes and subcloned in the modified pCHMWS lentiviral vector (kind gift from V. Baekelandt, University of Leuven, Belgium) sites by removing the fLUC region. The correct sequence was verified by sequencing. For *NF1* silencing, sh*NF1* from pLKO (Sigma, TRCN0000238778) (sh*NF1*_1) was subcloned in pGIPZ lentiviral vector (Open Biosystems). The corresponding short hairpin sequence was synthetized (GATC) and amplified by PCR using XhoI and EcoRI sites containing primers. The PCR product was digested using XhoI and EcoRI and subcloned into the pGIPZ vector previously digested with XhoI and PmeI followed by digestion with EcoRI. The two vector fragments were ligated with *NF1* short hairpin fragment. The correct insertion and sequence was validated by sequencing. In addition, experiments were performed using shNF1-pGIPZ clone V2LHS_76027 (sh*NF1*_4) and V2LHS_260806 (sh*NF1*_5).

RCAS viruses (RCAS-shNf1, RCAS-sgNf1, and RCAS-Kras$^{G12V}$) used for infection of p53-null NSCs were obtained from previously transfected DF1 chicken fibroblasts (ATCC #CRL-12203) using FuGENE 6 Transfection reagent (Promega #E2691), according to the manufacturer's protocol. DF1 cells were grown at 39°C in DMEM containing GlutaMAX (Gibco #31966-021) and 10% FBS (Sigma-Aldrich #F7524).

The pKLV-U6gRNA-PGKpuro2ABFP was a gift from Dr. Kosuke Yusa (Wellcome Sanger Institute) (Addgene plasmid #50946). For cloning of single gRNAs, oligonucleotides containing the BbsI site and the specific gRNA sequences were annealed, phosphorylated, and ligated into the pKLV-U6gRNA(BbsI)-PGKpuro2ABFP previously digested with BbsI. Single gRNAs to target *Fosl1* were designed with Guide Scan (http://www.guidescan.com/) and the sequences cloned were sg*Fosl1*_1: TACCGAGACTACGGGGAACC; sg*Fosl1*_2: CCTAGGGCTCGTATGACTCC; sg*Fosl1*_3: ACCG TACGGGCTGCCAGCCC. These vectors and a non-targeting sgRNA control were used to transduce p53-null *Kras*$^{G12V}$ NSCs.

The pLVX-Cre and respective control vector were kindly provided by Dr. Maria Blasco (CNIO) and used to transduce *Fosl1*$^{TetON}$ NSCs; pSIN-EF1-puro-FLAG-FOSL1 pLKO.1-TET-sh*FOSL1*_3 and sh*FOSL1*_10 and respective control vectors were a gift from Dr. Silve Vicent (CIMA, Navarra University); pBabe-FOSL1 was previously described (*Matsuo et al., 2000*).

Gp2-293 packaging cell line (Clontech #631458) was grown in DMEM (Sigma-Aldrich #D5796) with 10% FBS. Lentiviruses generated in this cell line were produced using calcium-phosphate precipitate transfection and co-transfected with second-generation packaging vectors (pMD2G and psPAX2). High-titer virus was collected at 36 and 60 hr following transfection.

All cells were infected with lenti- or retroviruses by four cycles of spin infection (200 × g for 2 hr) in the presence of 8 µg/mL polybrene (Sigma-Aldrich #H9268). Transduced cells were selected after 48 hr from the last infection with 1 µg/mL puromycin (Sigma-Aldrich #P8833).

## Generation of murine gliomas p53-null *Kras*$^{G12V}$

Fosl1$^{TetON}$ NSCs ($5 \times 10^5$ cells) were intracranially injected into 4- to 5-week-old wildtype C57Bl/6J female mice that were fed ad libitum with 2 g/kg doxycycline-containing pellets. Due to the limited penetration of the blood–brain barrier and to ensure enough Dox was reaching the brain, 2 mg/mL Dox (PanReac AppliChem #A29510025) was also added to drinking water with 1% sucrose (Sigma-Aldrich #S0389) (*Annibali et al., 2014*; *Mansuy and Bujard, 2000*). Control mice were kept with regular food and 1% sucrose drinking water.

Mice were anesthetized with 4% isofluorane and then injected with a stereotactic apparatus (Stoelting) as previously described (*Hambardzumyan et al., 2009*). After intracranial injection, all mice were routinely checked and sacrificed when developed symptoms of disease (lethargy, poor grooming, weight loss, and macrocephaly).

## Immunohistochemistry

Tissue samples were fixed in 10% formalin, paraffin-embedded, and cut in 3 µm sections, which were mounted in Superfrost Plus microscope slides (Thermo Scientific #J1810AMNZ) and dried. Tissues were deparaffinized in xylene and re-hydrated through graded concentrations of ethanol in water, ending in a final rinse in water.

For histopathological analysis, sections were stained with hematoxylin and eosin (H&E).

For immunohistochemistry, deparaffinized sections underwent heat-induced antigen retrieval, endogenous peroxidase activity was blocked with 3% hydrogen peroxide (Sigma-Aldrich #H1009)

for 15 min, and slides were then incubated in blocking solution (2.5% BSA [Sigma-Aldrich #A7906] and 10% Goat serum [Sigma-Aldrich #G9023], diluted in PBS) for at least 1 hr. Incubations with anti-FRA-1 (Santa Cruz #sc-183, 1:100) and anti-CD44 (BD Biosciences #550538, 1:100) were carried out overnight at 4°C. Slides were then incubated with secondary anti-rabbit (Vector #BA-1000) or anti-rat (Vector #BA-9400) for 1 hr at RT and with AB (avidin and biotinylated peroxidase) solution (Vectastain Elite ABC HRP Kit, Vector, PK-6100) for 30 min. Slides were developed by incubation with peroxidase substrate DAB (Vector SK-4100) until desired stain intensity. Finally, slides were counterstained with hematoxylin, cleared, and mounted with a permanent mounting medium.

Immunohistochemistry for S100A4 (Abcam #ab27957, 1:300) and Ki67 (Master Diagnostica #0003110QD, undiluted) was performed using an automated immunostaining platform (Ventana discovery XT, Roche).

## Immunoblotting

Cell pellets or frozen tumor tissues were lysed with JS lysis buffer (50 mM HEPES, 150 mM NaCl, 1% glycerol, 1% Triton X-100, 1.5 mM $MgCl_2$, 5 mM EGTA), and protein concentrations were determined by DC protein assay kit II (Bio-Rad #5000112). Proteins were separated on house-made SDS-PAGE gels and transferred to nitrocellulose membranes (Amersham #10600003). Membranes were incubated in blocking buffer (5% milk in TBST) and then with primary antibody overnight at 4°C. The following primary antibodies and respective dilutions were used: FLAG (Cell Signaling Technology #2368S, 1:2000), FRA-1 (Santa Cruz #sc-183, 1:1000; #sc-605, 1:1000), GFAP (Sigma-Aldrich #G3893, 1:5000), NF1 (Santa Cruz #sc-67, 1:500; Bethyl #A300-140A, 1:1000), OLIG2 (Millipore #AB9610, 1:2000), VIMENTIN (Cell Signaling Technology #5741, 1:3000), p-ERK1/2 (T202/Y204) (Cell Signaling Technology, #9101, 1:2000/3000; Assay Designs #ADI-905-651, 1:250), ERK1/2 (Cell Signaling Technology, #9102, 1:1000; Abcam #ab17942, 1:1000), p-MEK (S217/221) (Cell Signaling Technology, #9154, 1:500/1000), MEK (Cell Signaling Technology, #9122 1:1000), CHI3L1 (Qidel #4815, 1:1000), p85 (Millipore #06-195, 1:10,000), vinculin (Sigma-Aldrich #V9131, 1:10,000), and α-tubulin (Abcam #ab7291, 1:10,000). Anti-mouse or rabbit-HRP-conjugated antibodies (Jackson ImmunoResearch, #115-035-003 and #111-035-003) were used to detect desired protein by chemiluminescence with ECL Detection Reagent (Amersham, #RPN2106).

## Reverse transcription quantitative PCR

RNA from NSCs and frozen tissue was isolated with TRIzol reagent (Invitrogen #15596-026) according to the manufacturer's instructions. For reverse transcription PCR (RT-PCR), 500 ng of total RNA was reverse transcribed using the High Capacity cDNA Reverse Transcription Kit (Applied Biosystems #4368814). Quantitative PCR was performed using the SYBR Select Master Mix (Applied Biosystems #4472908) according to the manufacturer's instructions. qPCRs were run and the melting curves of the amplified products were used to determine the specificity of the amplification. The threshold cycle number for the genes analyzed was normalized to GAPDH. Mouse and human primer sequences are listed in *Supplementary file 6*.

RNA from BTSC cells was prepared using the RNeasy kit or the AllPrep DNA/RNA Protein Mini Kit and used for first-strand cDNA synthesis using random primers and SuperscriptIII reverse transcriptase (Life Technologies #18080-085). Primer sequences used in qRT-PCR with SYBR Green are listed in *Supplementary file 6*. Quantitative real-time PCR (qRT-PCR) STAT3 and CEBPB were performed using pre-validated TaqMan assays (Applied Biosystems): STAT3: Hs01047580, CEBPB: Hs00270923 and 18S rRNA: Hs99999901.

## MTT assay

Cells were seeded in 96-well culture plates (1000 cells per well, 10 wells per cell line) and grown for 7 days. At each timepoint (days 1, 3, 5, and 7), cell viability was determined by MTT assay. Briefly, 10 μL of 5 mg/mL MTT (Sigma-Aldrich #M5655) was added to each well and cells were incubated for 4 hr before lysing with a formazan solubilization solution (10% SDS in 0.01 M HCl). Colorimetric intensity was quantified using a plate reader at 590 nm. Values were obtained after subtraction of matched blanks (medium only).

## Cell cycle analysis: propidium iodide (PI) staining

Cells were harvested and washed twice with PBS prior to fixation with 70% cold ethanol, added drop-wise to the cell pellet while vortexing. Fixed cells were then washed, first with 1% BSA in PBS, then with PBS only and stained overnight with 50 µg/mL PI (Sigma-Aldrich #P4170) and 100 µg/mL RNase A (Roche #10109142001) in PBS. Samples were acquired in a FACSCanto II cytometer (BD Biosciences), and data were analyzed using FlowJo software.

## BrdU and EdU incorporation assays

Cells were pulse-labeled with 10 µM BrdU (Sigma-Aldrich #B9285) for 2 hr, harvested and washed twice with PBS prior to fixation with 70% cold ethanol, and added drop-wise to the cell pellet while vortexing. DNA denaturation was performed by incubating samples for 10 min on ice with 0.1 M HCl with 0.5% Tween-20, and samples were then resuspended in water and boiled at 100°C for 10 min. Anti-BrdU-FITC antibody (BD Biosciences #556028) was incubated according to the manufacturer's protocol. After washing with PBSTB (PBS with 0.5% Tween-20% and 1% BSA), samples were resuspended in 25 µg/mL PI and 100 µg/mL RNase A diluted in PBS. Samples were acquired in a FACSCanto II cytometer (BD Biosciences), and data were analyzed using FlowJo software.

EdU incorporation was assessed using the EdU-Click594 Cell Proliferation Imaging Kit (Baseclick GmbH) according to the manufacturer's instructions. 96 hr after transduction, $2.0 \times 10^4$ BTSC 233 cells were seeded on laminin-coated glass coverslips in a 24-well cell culture plate. Pictures were acquired using an Axiovert Microscope (Zeiss).

## Immunofluorescence

Cells were plated in laminin-coated coverslips and fixed with 4% PFA for 15 min. Cells were then permeabilized with 0.1% Triton X-100 in 0.2% BSA, and coverslips were washed and blocked with 10% donkey serum in 0.2% BSA for 1 hr. The following primary antibodies were incubated overnight at 4°C: CD44 (BD Biosciences #550538, 1:100), GFAP (Millipore #MAB360, 1:400), and OLIG2 (Millipore #AB9610, 1:100). Secondary antibodies at 1:400 dilution (from Invitrogen, Alexa-Fluor anti-rabbit-488, anti-mouse-488, and anti-rat 594) were incubated for 1 hr at RT and after washing coverslips were incubated for 4 min with DAPI (1:4000, Sigma-Aldrich #D8417) and mounted with ProLong Gold Antifade reagent (Invitrogen #P10144).

Fluorescence signal was quantified as the ratio of green/red pixel area relative to DAPI pixel area per field of view in a total of 36 fields per condition analyzed.

## Neurosphere formation assay and limiting dilution analysis

Neurospheres were dissociated and passed through a 40 µm mesh filter to eliminate non-single cells. Decreasing cell densities were plated in ultra-low attachment 96-well plates (Corning #CLS3474), and fresh medium was added every 3–4 days. The number of positive wells for the presence of spheres was counted 2 weeks after plating. Limiting dilution analysis was performed using ELDA R package (http://bioinf.wehi.edu.au/software/elda/).

## RNA-sequencing and analysis on mouse NSCs

For the p53-null $Kras^{G12V}$ NSCs, 1 µg of total RNA from the samples was used. cDNA libraries were prepared using the 'QuantSeq 3 'mRNA-Seq Library Prep Kit (FWD) for Illumina' (Lexogen #015) by following the manufacturer's instructions. Library generation is initiated by reverse transcription with oligo(dT) priming, and a second-strand synthesis is performed from random primers by a DNA polymerase. Primers from both steps contain Illumina-compatible sequences. Adapter-ligated libraries were completed by PCR, applied to an Illumina flow cell for cluster generation, and sequenced on an Illumina HiSeq 2500 by following the manufacturer's protocols. Sequencing read alignment and quantification and differential gene expression analysis was performed in the Bluebee Genomics Platform, a cloud-based service provider (https://www.illumina.com/company/about-us/mergers-acquisitions/bluebee.html). Briefly, reads were first trimmed with bbduk from BBTools (BBMap – Bushnell B, https://sourceforge.net/projects/bbmap/) to remove adapter sequences and polyA tails. Trimmed reads were aligned to the GRCm38/mm10 genome assembly with STAR v 2.5 (*Dobin et al., 2013*). Read counting was performed with HTSeq (*Anders et al., 2015*). The list of

stem/differentiation markers was compiled by combining a previously described gene list (*Sandberg et al., 2013*) with other markers (*Bazzoli et al., 2012*).

For the p53-null sh*Nf1* NSCs, total RNA samples (500 ng) were converted into sequencing libraries with the 'NEBNext Ultra II Directional RNA Library Prep Kit for Illumina' (NEB #E7760), as recommended by the manufacturer. Briefly, polyA+ fraction is purified and randomly fragmented, converted to double-stranded cDNA, and processed through subsequent enzymatic treatments of end-repair, dA-tailing, and ligation to adapters. Adapter-ligated library is completed by PCR with Illumina PE primers. The resulting purified cDNA libraries were applied to an Illumina flow cell for cluster generation and sequenced on an Illumina NextSeq 550 by following the manufacturer's protocols. We then used the Nextpresso pipeline (*Graña et al., 2018*) for alignment and quantification.

## Osteogenesis differentiation assay

The osteogenesis differentiation assay was performed using the StemPro Osteogenesis Differentiation Kit (Life Technologies #A1007201) according to the manufacturer's instructions. Briefly, $5 \times 10^3$ cells/cm$^2$ were seeded on laminin-coated glass coverslips in a 24-well cell culture plate. Cells were incubated in MSC Growth Medium at 37°C, 5% $CO_2$ for 21 days, replacing the medium every 4 days. Cells were then fixed with 4% formaldehyde, stained with Alizarin Red S solution (pH 4.2), and mounted on microscope slides. Pictures were acquired using an Axiovert Microscope (Zeiss).

## Active Ras pull-down assay

Active Ras pull-down assay was performed using Active Ras pull-down assay kit (Thermo Fisher Scientific #16117) according to the manufacturer's instructions. Briefly, cells were grown in 10 cm plates at 80–90% confluency in the presence or absence of growth factors (EGF, FGF, and LIF) and lysed with the provided buffer. Lysates were incubated with either GDP or GTP for 30 min followed by precipitation with GST-Raf1-RBD. Western blot was performed with the provided anti-RAS antibody (1:250).

## Chromatin preparation and FRA-1 ChIP

BTSC cells were collected at $2 \times 10^6$ cells confluency, washed in PBS, and fixed by addition of 1% formaldehyde for 20 min at room temperature. The cells were resuspended in 2 mL lysis buffer (50 mM Tris pH 7.5; 1 mM EDTA pH 8.0; 1% Triton; 0.1% Na-deoxycholate; 150 mM NaCl; protease inhibitors) on ice for 20 min. The suspension was sonicated in a cooled Bioruptor Pico (Diagenode) and cleared by centrifugation for 10 min at 13,000 rpm. The chromatin (DNA) concentration was quantified using NanoDrop (Thermo Scientific), and the sonication efficiency was monitored on an agarose gel. Protein A/G plus-agarose beads (Santa Cruz #sc-2003) were blocked with sonicated salmon sperm (Thermo Fisher #AM9680, 200 mg/mL beads) and BSA (NEB #B9000S, 250 mg/mL beads) in dilution buffer (0.5% NP40; 200 mM NaCl; 50 mM Tris pH 8.0; protease inhibitors) for 2 hr at room temperature. The chromatin was pre-cleared with 80 µL of blocked beads for 1 hr at 4°C. Pre-cleared chromatin was incubated with 5 µg of FRA-1 antibody (Santa Cruz #sc-605) overnight at 4°C, then with 40 µL of blocked beads for further 2 hr at 4°C. Control mock immunoprecipitation was performed with blocked beads. The beads were washed 1× with Wash1 (20 mM Tris pH 7.5; 2 mM EDTA pH 8.0; 1% Triton; 0.1% SDS; 150 mM NaCl), 1× with Wash2 (20 mM Tris pH 7.5; 2 mM EDTA pH 8.0; 1% Triton; 0.1% SDS; 500 mM NaCl), 1× with LiCl Wash (20 mM Tris pH 7.5; 1 mM EDTA pH 8.0; 1% NP40; 1% deoxycholate; 250 mM LiCl), and 2× in TE (10 mM Tris pH 7.5; 1 mM EDTA). The immuno-complexes were eluted by two 15 min incubations at 30°C with 100 µL Elution buffer (1% SDS, 100 mM NaHCO$_3$), and de-crosslinked overnight at 65°C in the presence of 10 U RNase A (Ambion #AM9780). The immune-precipitated DNA was then purified with the QIAquick PCR purification kit (QIAGEN #28104) according to the manufacturer's protocol and analyzed by quantitative real-time PCR.

## Statistical analysis

All statistical analyses were performed using R programming language (3.6.3). Statistical differences between groups were assessed by one-way ANOVA, two-way ANOVA, or unpaired two-tailed Student's t tests unless otherwise specified.

Kaplan–Meier survival curves were produced with GraphPad Prism, and p-values were generated using the log-rank statistics.

Results are presented as mean ± standard deviation (SD), and statistical significance was defined as p≤0.05 for a 95% confidence interval.

## Code availability

The accession numbers for data reported in this paper are GEO: GSE137310 (Freiburg BTSCs) and GSE138010 (mouse p53-null *Kras^{G12V}* NSCs). All the R code and data used for analysis and plots generation are available at: https://github.com/squatrim/Marques2020 [copy archived at swh:1:rev:45e31e7d17f006d2d3a17e66a63449f758bf5998 (*Squatrito, 2021*)].

## Acknowledgements

We thank Álvaro Ucero for his input on the project and Flora A Díaz for her technical support. We are grateful to Francisco X Real and Scott Lowe for critical input on the manuscript. We thank Pamela Franco for experimental support and discussion. We are grateful to Peter Dirks and Trevor Pugh for providing the processed bulk RNA-seq data of their GSCs dataset. CM was supported by a 'La Caixa' predoctoral fellowship. YD was supported by the Berlin School of Integrative Oncology (BSIO) of the Berlin Charité Medical University. The GG lab acknowledges funding from MDC, Helmholtz (VH-NG-1153), and ERC (714922). This work was supported by a grant from the Marie Curie International re-integration Grants (MC-IRG), project no. 268303 (to MSC), and by grants from the ISCIII, project PI13/01028, cofounded by the European Regional Development Fund (ERDF), and from the Seve Ballesteros Foundation (to MS).

## Additional information

### Funding

| Funder | Grant reference number | Author |
|---|---|---|
| La Caixa Foundation | | Carolina Marques |
| Berlin School of Integrative Oncology, Charité – Universitätsmedizin Berlin | | Yuliia Dramaretska |
| MDC | VH-NG-1153 | Gaetano Gargiulo |
| European Research Council | 714922 | Gaetano Gargiulo |
| Marie CurieInternational re-integration Grants | 268303 | Maria Stella Carro |
| Instituto de Salud Carlos III | PI13/01028 | Massimo Squatrito |
| Seve Ballesteros Foundation | | Massimo Squatrito |

The funders had no role in study design, data collection and interpretation, or the decision to submit the work for publication.

### Author contributions

Carolina Marques, Formal analysis, Investigation, Methodology, Writing - original draft; Thomas Unterkircher, Paula Kroon, Barbara Oldrini, Annalisa Izzo, Roberto Ferrarese, Eva Kling, Investigation; Yuliia Dramaretska, Investigation, Visualization, Writing - original draft; Oliver Schnell, Sven Nelander, Resources; Erwin F Wagner, Latifa Bakiri, Conceptualization, Writing - review and editing; Gaetano Gargiulo, Conceptualization, Formal analysis, Supervision, Funding acquisition, Visualization, Writing - review and editing; Maria Stella Carro, Conceptualization, Resources, Data curation, Formal analysis, Supervision, Funding acquisition, Writing - original draft; Massimo Squatrito, Conceptualization, Resources, Data curation, Formal analysis, Supervision, Funding acquisition, Visualization, Methodology, Writing - original draft

Author ORCIDs
Carolina Marques (iD) https://orcid.org/0000-0002-8308-5630
Sven Nelander (iD) http://orcid.org/0000-0003-1758-1262
Maria Stella Carro (iD) https://orcid.org/0000-0002-8570-7691
Massimo Squatrito (iD) https://orcid.org/0000-0002-4593-3790

### Ethics

Animal experimentation: Patient-derived glioblastoma stem cells (BTSCs) from Freiburg University were prepared from tumor specimens under IRB-approved guidelines (# 407/09_120965). Mice were housed in the specific pathogen-free animal house of the Spanish National Cancer Research Centre under conditions in accordance with the recommendations of the Federation of European Laboratory Animal Science Associations (FELASA). All animal experiments were approved by the Ethical Committee (CEIyBA) (# CBA 31_2019-V2) and performed in accordance with the guidelines stated in the International Guiding Principles for Biomedical Research Involving Animals, developed by the Council for International Organizations of Medical Sciences (CIOMS).

### Decision letter and Author response

Decision letter https://doi.org/10.7554/eLife.64846.sa1
Author response https://doi.org/10.7554/eLife.64846.sa2

## Additional files

### Supplementary files

- Source code 1. R programming code used for data analysis and plot generation.
- Source data 1. Raw images for western blots.
- Supplementary file 1. Transcriptional subtypes of the brain tumor stem cell (BTSC) lines.
- Supplementary file 2. Genes differentially expressed in brain tumor stem cells (BTSCs) (mesenchymal [MES] versus non-MES) at FDR < 0.05.
- Supplementary file 3. Master regulator analysis (MRA) results.
- Supplementary file 4. TCGA and CGGA data.
- Supplementary file 5. Gene signatures used for the gene set enrichment analysis (GSEA).
- Supplementary file 6. Primers used in this study.
- Transparent reporting form

### Data availability

The accession numbers for data reported in this paper are GEO: GSE137310 (Freiburg BTSCs) and GSE138010 (mouse p53-null KrasG12V NSCs). All the R code and data used for analysis and plots generation is available at: https://github.com/squatrim/Marques2020 (copy archived at https://archive.softwareheritage.org/swh:1:rev:45e31e7d17f006d2d3a17e66a63449f758bf5998).

The following datasets were generated:

| Author(s) | Year | Dataset title | Dataset URL | Database and Identifier |
|---|---|---|---|---|
| Unterkircher T, Carro MS | 2019 | Genome-wide analysis of GBM-derived brain tumor stem cells-like (BTSCs) | https://www.ncbi.nlm. nih.gov/geo/query/acc. cgi?acc=GSE137310 | NCBI Gene Expression Omnibus, GSE137310 |
| Squatrito M | 2020 | Fosl1 regulates mesenchymal GBM plasticity | https://www.ncbi.nlm. nih.gov/geo/query/acc. cgi?acc=GSE138010 | NCBI Gene Expression Omnibus, GSE138010 |

The following previously published datasets were used:

| Author(s) | Year | Dataset title | Dataset URL | Database and Identifier |
|---|---|---|---|---|
| Squatrito M | 2019 | CHROMATIN LANDSCAPES REVEAL DEVELOPMENTALLY ENCODED TRANSCRIPTIONAL STATES THAT DEFINE GLIOBLASTOMA | https://www.ncbi.nlm.nih.gov/geo/query/acc.cgi/GSE119834 | NCBI Gene Expression Omnibus, GSE119834 |
| Squatrito M | 2015 | Precursor States of Brain Tumor Initiating Cell Lines Are Predictive of Survival in Xenografts and Associated With Glioblastoma Subtypes | https://trace.ddbj.nig.ac.jp/DRASearch/study?acc=SRP057855 | DRASearch, SRP057855 |
| Squatrito M | 2015 | Analysis of mRNA profiles distinguishes proneural (PN) glioma stem cells (GSC) from mesenchymal (Mes) GSCs | https://www.ncbi.nlm.nih.gov/geo/query/acc.cgi?acc=GSE67089 | NCBI Gene Expression Omnibus, GSE67089 |
| Squatrito M | 2007 | Expression analyses of glioblastoma derived neurosphere cultures | https://www.ncbi.nlm.nih.gov/geo/query/acc.cgi?acc=GSE8049 | NCBI Gene Expression Omnibus, GSE8049 |
| Squatrito M | 2013 | A Proneural to Mesenchymal Transition Mediated by NFkB Promotes Radiation Resistance in Glioblastoma (part 1) | https://www.ncbi.nlm.nih.gov/geo/query/acc.cgi?acc=GSE49161 | NCBI Gene Expression Omnibus, GSE49161 |
| Squatrito M | 2021 | Gradient of Developmental and Injury Response transcriptional states defines functional vulnerabilities underpinning glioblastoma heterogeneity. | https://singlecell.broad-institute.org/single_cell/study/SCP503/ | SCP503, single_cell |
| Squatrito M | 2019 | An Integrative Model of Cellular States, Plasticity, and Genetics for Glioblastoma. | https://singlecell.broad-institute.org/single_cell/study/SCP393/ | single_cell, SCP393 |

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
