## [Decision Letter]

**Acceptance summary:**

In this study, authors report a mechanism to link NF1/RAS/MAPK signalling and the acquisition of a MES gene expression program. Furthermore, they identify the AP-1 transcription factor FOSL1 as a central regulator of the glioma-intrinsic MES gene expression program, downstream of NF1 signalling. This study defines a direct mechanistic link between NF1-MAPK signalling and MES glioblastoma and recognizes that FOSL1 is a key transcription factor, downstream of NF1, that modulates glioblastoma stem cell plasticity.

**Decision letter after peer review:**

Thank you for submitting your article "NF1 regulates mesenchymal glioblastoma plasticity and aggressiveness through the AP-1 transcription factor FOSL1" for consideration by *eLife*. We apologize for delay in getting back to you. We had to contact many reviewers before we were able to engage 3 and unfortunately, the third reviewer never got back to us, so we are sending you a decision based on the two reviews that we have received, one of whom is a member of our Board of Reviewing Editors. The evaluation has been overseen Erica Golemis as the Senior Editor. The reviewers have opted to remain anonymous.

Essential Revisions:

1. Please use a preset cutoff for gene expression analyses. For example, Figure 1f uses unbalanced groups. Using a median expression level shows that the CGGA is not prognostic and the TCGA has a less significant result. The selective use of mRNA expression levels may also alter other results.

2. The use of a mutant Kras to induce tumors is not clearly explained. While Hras and Kras have been used in older GEMMs, the lack of oncogenic mutations in most gliomas raises concerns about how representative these studies are. The NF1 modeling is much better and should be used in place of Kras.

3. Are the changes in cell-cycle profiles upon KD of FOSL1 comparable between NGSs and BTSCs? Cell cycle profiling should be done in the BTSCs as well. Please validate the role of FOSL1 in proneural and mesenchymal patient-derived cells in functional studies.

4. Please validate the FOSL1 ChIP in glioma cells instead of HCT116. It seems less relevant to have these studies in a completely different cancer type.

5. The connection to stemness seems less direct. In Figure 7e, FOSL1 can alter cell growth without a direct connection to stemness. It would be helpful to test FOSL1 dependence in stem cells and differentiated cells.

6. In Figure 8, it would be helpful to directly test the effects of inhibiting the RAS-MAPK-ERK pathway, perhaps through inhibitors. The role of SRF/ELK is not really addressed.

7) Does overexpression of NF1 (NF1-GRD) in patient-derived BTSCs phenocopy the effect of FOSL1 deletion on the cell growth, cell cycle profiles, self-renewal and tumorigenicity.

---

## [Author Response]

Essential Revisions:1. Please use a preset cutoff for gene expression analyses. For example, Figure 1f uses unbalanced groups. Using a median expression level shows that the CGGA is not prognostic and the TCGA has a less significant result. The selective use of mRNA expression levels may also alter other results.

The cutoffs normally used for survival analysis are arbitrary. As a convention, it is often used either (a) the medium gene expression (50% vs 50% of the population), or (b) high/low vs the rest of the population (e.g. 30% vs 70%) or (c) high vs low (e.g. 30% vs 30%). For our original analysis we had compared the 30% high FOSL1 patients vs the rest of the population (the option b above) and that explains the two unbalanced groups. The reasoning behind the cutoff selection was to understand whether the patients with the highest level of expression of FOSL1 had the worst prognosis as compared to the rest of the patients. The data of both the TCGA and the CGGA suggested so. In order to have two balanced groups in the revised versions we have decided to compare the 30% of the patients with the highest FOSL1 expression vs the 30% with the lowest expression, one of the other commonly used cutoffs. The statistical significance of this latest analysis is even higher as compared to the original one, confirming indeed that FOSL1 high levels of expression are associated with the worst prognosis in IDHwt gliomas. Actually, while we were repeating the survival analysis, we realized that for the CGGA dataset we had erroneously included only the subset of IDHwt GBM. The error has been corrected in the revised analysis. For full transparency, we also include in this rebuttal letter the analysis performed either using the medium expression or the high vs the rest of the population.

**Author response image 1. sa2fig1:** Comparison of different cutoff groups used for the survival analysis.

2. The use of a mutant Kras to induce tumors is not clearly explained. While Hras and Kras have been used in older GEMMs, the lack of oncogenic mutations in most gliomas raises concerns about how representative these studies are. The NF1 modeling is much better and should be used in place of Kras.

We acknowledge that additional justification for the use of a KRas mutant over a Nf1 loss mouse model is in order. Ras activating mutations, in combination with other alterations as Akt mutation, loss of Ink4a/Arf or Trp53 have been widely used to study gliomagenesis (Friedmann-Morvinski et al., 2012; Holland et al., 2000; Koschmann et al., 2016; Muñoz et al., 2013; Uhrbom et al., 2002) and Kras mutant patients are rare but may be found in TCGA. Of note, Friedmann-Morvinski had previously shown that mouse models of gliomas induced by Ras mutants closely phenocopy loss of Nf1 (Author response image 2). Our in vitro experiments using the p53 null-NSCs (Figure 4) had evidenced that, as compared to loss of Nf1, Kras mutants induced a stronger upregulation of Fosl1 and of the other mesenchymal genes. Therefore, we have decided to use the Kras mutant model in order to properly address the role of Fosl1 in the MAPK-induced mesenchymal gliomas. To make this more explicit, at the end of page 8 we have now included the following comment:

“Altogether, these data indicated that KrasG12V–transduced cells, that show the highest FOSL1 expression and mesenchymal commitment, are a suitable model to functionally study the role of a MAPKFOSL1 axis in MES GBM.”

Moreover, in this revised version of our manuscript we have now knocked-out Fosl1 also in the p53-null NSCs transduced with the shNf1. As shown in Figure 2—figure supplement 3E-G, and similar to what observed for the Kras mutant model, RNA-seq and GSEA analysis suggest that loss of Fosl1 lead to a MES-PN transition also in the p53-null shNf1 NSCs. These data further support the idea that Kras mutants and Nf1 loss phenocopy each other, at least in the models that we have used for our studies.

**Author response image 2. sa2fig2:** Glioma generated with Ras mutants closely resemble those generated by Nf1 loss (image from Friedmann-Morvinski et al., 2012).

3. Are the changes in cell-cycle profiles upon KD of FOSL1 comparable between NGSs and BTSCs? Cell cycle profiling should be done in the BTSCs as well. Please validate the role of FOSL1 in proneural and mesenchymal patient-derived cells in functional studies.

In figure 7C we show the BrdU assay, which well documented proliferation defects.

In response to this reviewer’s request, we have performed the quantification of the cell-cycle profiles in the BTSC380 (Author response image 3). Consistently with what we observed in the mouse NSCs, and with the reduced incorporation of BrdU, silencing of FOSL1 led to a reduction of the fraction of cells in S-phase, albeit the two shRNAs used to silence FOSL1 differentially affect G1 or G2. Overall, together with the other experiments, this also indicates that loss of FOSL1 affects cell proliferation.

**Author response image 3. sa2fig3:** Cell cycle profiles of BTSC 380 upon *FOSL1* silencing.

4. Please validate the FOSL1 ChIP in glioma cells instead of HCT116. It seems less relevant to have these studies in a completely different cancer type.

FOSL1/FRA-1 ChIP experiments performed on the BTSC349 cell were already included in the original manuscript. Please check Figure 7J.

5. The connection to stemness seems less direct. In Figure 7e, FOSL1 can alter cell growth without a direct connection to stemness. It would be helpful to test FOSL1 dependence in stem cells and differentiated cells.

The limiting dilution experiments are the gold-standard assays used to measure stemness of glioma stem-like cells (see for reference Seyfrid M et al., 2019; https://doi.org/10.1007/978-1-4939-8805-1_7). Our experiments, both in mouse NSCs (Figure 5C) and human BTSCs (Figure 7E) clearly demonstrate that loss of FOSL1 leads to a significant reduction in the stem cell frequency. In order to further address the role of FOSL1 in differentiated cells we induced the differentiation of BTSC380, by growing them in neurosphere medium + 0.5%FBS and 5ng/ml TNFalpha for 5 days (Figure 7—figure supplement 1F). Similar to what is observed in stemness condition (Figure 7C), silencing of FOSL1 is associated with drastic reduction in proliferation also in differentiated cells. These data indicate that mesenchymal glioma cells, both stem and differentiated, depend on FOSL1 for their growth.

6. In Figure 8, it would be helpful to directly test the effects of inhibiting the RAS-MAPK-ERK pathway, perhaps through inhibitors. The role of SRF/ELK is not really addressed.

As previously shown in Figure S4 and now in Figure 2—figure supplement 3, inhibition of the RASMAPK-ERK pathway with 3 independent inhibitors, both in human BTSCs (Figure 2-figure supplement 3A-B) and mouse NSCs (Figure 2—figure supplement 3C-D) led to a reduction of both FOSL1 as well as of other mesenchymal genes. This data clearly indicated that the activation of the MAPK signaling downstream of the loss of NF1 is responsible for the induction of FOSL1 expression and in turn the activation of the mesenchymal gene signature. Regarding the role of SRF/ELK in the regulation of FOSL1 in mesenchymal gliomas, we agree with the reviewers that the data we had originally included were superficial and did not properly address the issue. Considering that the role of SRF/ELK is not an integral part of our studies we have decided to remove those data, including those in the proposed model in Figure 8, and we have maintained in the Discussion section only a comment regarding FOSL1 regulation by SRF.

7) Does overexpression of NF1 (NF1-GRD) in patient-derived BTSCs phenocopy the effect of FOSL1 deletion on the cell growth, cell cycle profiles, self-renewal and tumorigenicity.

In order to address this comment, we have measured EdU incorporation and the proliferation of BTSC233 transduced with the NF1-GRD (Figure 2—figure supplement 1E-F). Re-establishing NF1 expression in those cells leads to reduced EdU incorporation, with consequent reduction in the proliferation capacity. Although it might be an interesting question, the effect of NF1-GRD on the tumorigenicity of the BTSCs is out of the scope of our current study.